# GOLD PANNING: TURNING POSITIONAL BIAS INTO SIGNAL FOR MULTI-DOCUMENT LLM REASONING

## ABSTRACT

Large language models exhibit a strong position bias in multi-document contexts, systematically prioritizing information based on location rather than relevance. While existing approaches treat this bias as noise to be mitigated, we introduce GOLD PANNING BANDITS, a framework that leverages position bias as a diagnostic signal: by reordering documents and observing shifts in the model's responses, we can efficiently identify the most relevant content. We frame the problem of choosing reorderings as a bipartite matching problem. While an optimal assignment can be computed at each iteration with the Hungarian algorithm in $O(N^3)$ time, we propose a greedy $O(N \log N)$ strategy that achieves comparable performance by prioritizing the placement of the most uncertain documents in the most informative positions. Our approach identifies relevant documents using up to 65% fewer language model queries than random permutation baselines on knowledge-intensive NLP tasks, substantially reducing computational cost without model retraining. This work demonstrates that inherent LLM biases can be transformed from liabilities into assets for efficient, inference-time optimization.

## 1 INTRODUCTION

Knowledge-intensive tasks, ranging from legal discovery and literature review to multi-source analysis in science and policy, require identifying relevant information from large document collections. Large language models (LLMs) have emerged as powerful tools for this, effectively extracting and synthesizing knowledge across vast textual corpora. These capabilities are critical in retrieval-augmented generation (Lewis et al., 2020; Karpukhin et al., 2020; Gao et al., 2023) and agentic systems (Wang et al., 2023a; Yao et al., 2023; Shinn et al., 2023), which must process multiple pieces of information to support reliable decision-making. However, a key challenge undermining this reliability is *position bias*, where LLMs often rely more on *where* information appears in their context than on its intrinsic relevance (Wang et al., 2023b; Zheng et al., 2024; Liu et al., 2024). This shortcoming demands new approaches to information presentation that ensure relevant information receives fair consideration regardless of its position.

Even state-of-the-art retrieval systems (Izacard et al., 2022; Sarthi et al., 2024; Ke et al., 2025; Zhang et al., 2025), which rank documents by perceived relevance, may inadvertently bury critical information. To counteract this, existing approaches like Permutation Self-Consistency (PSC) (Tang et al., 2024) treat position bias as noise to be averaged away. PSC *randomly* shuffles documents across multiple inference passes and aggregates the results, ensuring that each document is exposed to different positions. While this ensures fairness in expectation, it is inefficient: *each shuffle is treated independently*, ignoring what was already learned about a document's relevance. Rather than discarding these observations, we can treat each permutation as an experiment that reveals information about document relevance: some positions act like sensitive detectors, surfacing even moderately relevant content, while others are less informative. This motivates the central problem of this work: finding an optimal strategy for permuting documents to most efficiently identify relevant information. We formalize this as a sequential assignment problem in §2.

Answering this question requires navigating several interconnected challenges. The space of possible document orderings grows factorially, making exhaustive evaluation impossible, while each query reveals information about only the specific ordering tested. Most critically, we face a fundamental tension between exploiting orderings we believe will surface relevant content and exploring

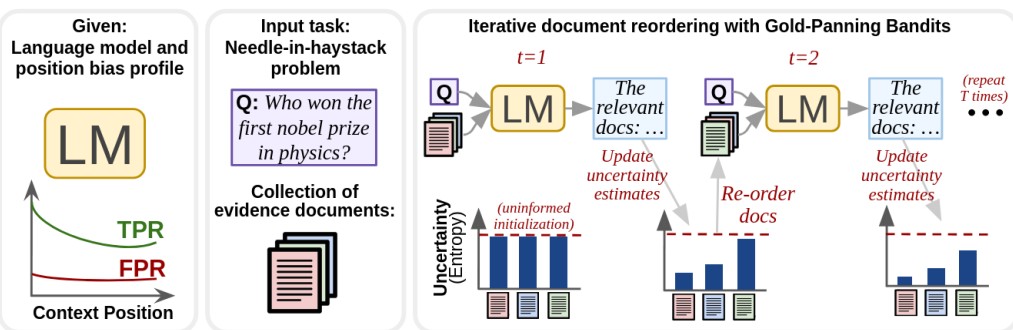

Figure 1: Overview of our GOLD PANNING algorithm. We leverage an LLM's known positional bias (left) to solve a needle-in-haystack task (center). Our iterative method (right) involves querying the model, updating beliefs about document relevance, and strategically reordering documents for the next query. By placing uncertain documents in the most informative positions, **we rapidly identify relevant content with fewer queries.**

new configurations that might reveal overlooked documents. This structure, sequential decisions under uncertainty with partial feedback and combinatorial action spaces, naturally leads us to formulate document permutation as a bandit problem (Slivkins, 2019).

To address this, we introduce GOLD PANNING BANDITS, a new class of combinatorial bandits (Gai et al., 2010; 2012; Chen et al., 2013) that fundamentally reconceptualizes position bias—transforming it from noise to be averaged away into a counterintuitive feature for efficient information discovery. To our knowledge, this is the first work to demonstrate that systematic LLM biases can be exploited rather than mitigated for inference-time optimization. Our key insight is that different positions offer different *lenses* through which to view documents, with some being highly biased and others more selective, creating unexplored opportunities for strategic information gathering that can dramatically reduce the number of LLM queries needed. Moreover, we are the first to formalize this as a bandit problem for long-context inference-time reasoning, where each query reveals information about only the specific ordering tested, and we must balance exploration of new configurations with exploitation of promising orderings.

Building on this framework, we propose GOLD PANNING, an efficient algorithm that for the first time employs a greedy strategy to actively leverage position bias for document relevance discovery. As illustrated in Figure 1, our method leverages a pre-calibrated positional bias profile of the LLM, where each context position acts as a detector, characterized by its **True Positive Rate** (TPR, the ability to surface relevant documents), and its **False Positive Rate** (FPR, its tendency to mistakenly highlight irrelevant ones). We contend that acquiring this profile is a lightweight, one-time procedure whose cost is amortized over all subsequent uses, making it fair to assume as a given input. With this profile, the algorithm iteratively updates its probabilistic belief about each document's relevance. At each step, it strategically places the most uncertain documents into the most informative positions to maximize information gain per query. The value of this upfront calibration is critical; an ablation study using Thompson Sampling to learn these TPR/FPR values online performs no better than random shuffling. In contrast, by leveraging this modest, one-time calibration, GOLD PANNING quickly homes in on relevant content. Through simulation and empirical validation, we demonstrate our method identifies relevant documents using up to 65% fewer model queries than random permutation strategies (e.g., reducing the number of passes from 20 to 7 in typical scenarios).

**Our primary contribution is demonstrating, for the first time, that LLM position bias can be exploited rather than mitigated**, transforming a known limitation into an algorithmic advantage. Building on this insight, we also contribute: **(a)** We introduce GOLD PANNING BANDITS, a new class of combinatorial bandits for modeling document permutation under position bias, **representing the first application of bandit algorithms to long-context inference-time reasoning**; **(b)** We design GOLD PANNING, an efficient greedy inference-time algorithm that solves GOLD PANNING BANDITS problems; **(c)** We provide theoretical analysis of GOLD PANNING's convergence and information-theoretic guarantees; **(d)** We validate our method on simulations and real-world tasks, showing up to 65% fewer required queries at comparable accuracy.

## 2 GOLD PANNING BANDITS: EXPLOITING POSITION BIAS

As motivated in the introduction, we now formalize the problem of strategically reordering documents to exploit position bias. Unlike traditional bandits where an arm's reward is an arm-specific constant (albeit unknown), here the value of pairing item $i$ with detector $j$ is the expected information gain, which depends jointly on the item's current belief *and* the detector's performance. This dynamic, belief-dependent utility distinguishes GOLD PANNING BANDITS from traditional combinatorial bandits, as we detail further in §6. We generalize from documents (items) and context positions (detectors) to develop a framework applicable to situations where objects with unknown states must be tested under heterogeneous settings.

We develop the GOLD PANNING BANDITS framework in three stages. We begin with the formal problem definition (§2.1), introducing items with hidden relevance states, detectors with heterogeneous diagnostic capabilities characterized by TPR/FPR pairs, and the assignment constraint that enforces one-to-one matchings between items and detectors. Next, we establish the Bayesian machinery for learning from observations (§2.2), deriving how binary detection outcomes update our beliefs about item relevance and defining an information-theoretic objective that quantifies the expected value of each assignment. Finally, we analyze the computational challenges (§2.3), showing why exact optimization is intractable and motivating the need for efficient approximation algorithms.

### 2.1 FORMAL PROBLEM DEFINITION

**Items and hidden states.** We consider $N$ items indexed by $i \in \{1, \ldots, N\}$, each possessing an unknown binary state $Z_i \in \{0, 1\}$ that we aim to discover. For instance, in document retrieval, each document has an unknown relevance $Z_i$, where $Z_i = 1$ indicates the document contains information pertinent to the query. Our objective is to determine the complete state vector $\mathbf{Z} = (Z_1, \ldots, Z_N)$ with high confidence using minimal rounds of testing.

**Detectors and diagnostic capability.** To discover these hidden states, we employ $M$ heterogeneous detectors indexed by $j \in \{1, \ldots, M\}$. We characterize each detector's ability to distinguish between states using two key parameters:

- True Positive Rate: $\mathrm{TPR}_j = \Pr(\text{detector outputs } 1 \mid Z = 1)$,
- False Positive Rate: $\mathrm{FPR}_j = \Pr(\text{detector outputs } 1 \mid Z = 0)$.

The discriminative power, or *diagnosticity*, of detector $j$ is quantified by $d_{\mathrm{diag}}(j) = |\mathrm{TPR}_j - \mathrm{FPR}_j|$. This measure, coinciding with Youden's J-statistic (Peirce, 1884; Youden, 1950), captures how well a detector distinguishes between the two states. A detector with $d_{\mathrm{diag}}(j) = 0$ provides no information (outputs are independent of true state), while $d_{\mathrm{diag}}(j) = 1$ represents perfect discrimination.

**The assignment constraint.** At each round, we must form a one-to-one matching between items and detectors. For analytical clarity, we frame the problem as a symmetric $N$-to-$N$ assignment. Asymmetric cases where the number of items and detectors differ are handled as follows: **When Items Exceed Detectors ($N > M$):** The agent must first select which $M$ items to test. We model this by framing the action as a single $N$-to-$N$ assignment where the agent matches $N - M$ items to *dummy* detectors with $\mathrm{TPR} = \mathrm{FPR}$ (zero diagnosticity). This makes the choice of which items to leave untested an explicit part of the permutation. **When Detectors Exceed Items ($M > N$):** The agent must select which $N$ of the $M$ available detectors to use. This selection can be strategic (e.g., always choosing the $N$ most diagnostic detectors) or, in some applications, architecturally constrained (e.g., being required to use the first $N$ context positions in an LLM). This reduction lets us analyze the core problem as a balanced $N$-to-$N$ assignment without loss of generality while reflecting practical constraints.

### 2.2 BELIEF UPDATES AND INFORMATION GAIN

**Actions and observations.** Given our symmetric $N$-to-$N$ framing, an action at each round $t$ is a *permutation* $\sigma_t \in S_N$ (the set of permutations over $\{1, \ldots, N\}$ (Beachy & Blair, 2006)). This permutation defines a complete one-to-one mapping, where $\sigma_t(i) = j$ indicates that item $i$ is tested by detector $j$ in that round. When item $i$ is tested by detector $j$, we observe a binary outcome

$O_{ij} \in \{0, 1\}$. The likelihood of a positive outcome is a function of the item's true state and the detector's properties:

$$\Pr(O_{ij} = 1 \mid Z_i = z, \text{TPR}_j, \text{FPR}_j) = z \cdot \text{TPR}_j + (1 - z) \cdot \text{FPR}_j. \tag{1}$$

The detector's parameters, $(\text{TPR}_j, \text{FPR}_j)$, may be estimated via preliminary calibration experiments (§D). If a *selective* detector (high TPR, low FPR) yields a positive outcome for item $i$, the posterior for $Z_i = 1$ increases substantially; conversely, a negative outcome from a *permissive* detector provides only weak negative evidence. We assume detector parameters are stationary and that, conditional on $\mathbf{Z}$ and the assignment, outcomes across pairs within a round are independent. While this may not hold in all applications, such as in transformer-based models where attention mechanisms create dependencies between positions, our empirical results (§5 and §B) suggest it is a robust assumption, likely because position effects dominate over inter-position dependencies.

**Bayesian belief dynamics.** We maintain posterior beliefs $\mathbf{b}_t = (b_{t,1}, \ldots, b_{t,N})$ where $b_{t,i} = \Pr(Z_i = 1 \mid \mathcal{F}_{t-1})$ represents our belief that item $i$ has state 1 given all observations through time $t - 1$. Starting from an uninformative prior $b_{0,i} = 0.5$, beliefs evolve through Bayesian updating:

$$b_{t+1,i} = \frac{b_{t,i} \cdot \Pr(O_{ij} \mid Z_i = 1, \text{TPR}_j, \text{FPR}_j)}{b_{t,i} \cdot \Pr(O_{ij} \mid Z_i = 1, \text{TPR}_j, \text{FPR}_j) + (1 - b_{t,i}) \cdot \Pr(O_{ij} \mid Z_i = 0, \text{TPR}_j, \text{FPR}_j)}, \tag{2}$$

with $j = \sigma_t(i)$. Strong evidence from diagnostic detectors produces large belief shifts, while weak signals from uninformative detectors yield minimal updates.

**Information-theoretic objective.** At each round, we aim to select the action that maximizes expected information gain, i.e., the expected reduction in our uncertainty about the hidden states $\mathbf{Z}$.

Under the conditional-independence assumption within a round, the vector of observations is

$$\mathbf{O}_t = \{O_{i,\sigma_t(i)}\}_{i=1}^N, \tag{3}$$

and the total information gain decomposes as:

$$\sigma_t^\star = \arg \max_{\sigma_t \in S_N} \mathcal{I}(\mathbf{Z}; \mathbf{O}_t \mid \sigma_t, \mathbf{b}_t) = \arg \max_{\sigma_t \in S_N} \sum_{i=1}^N \mathcal{I}(Z_i; O_{i,\sigma_t(i)} \mid b_{t,i}). \tag{4}$$

Each term $\mathcal{I}(Z_i; O_{ij} \mid b_{t,i})$ represents the expected information gain from testing item $i$ with detector $j$. As shown in the closed-form expression below (derived in Appendix A), this formalizes the principle of state-coupled utility:

$$\mathcal{I}(Z_i; O_{ij} \mid b_{t,i}) = \mathcal{H}(\text{FPR}_j + b_{t,i}\Delta_j) - \left( b_{t,i}\mathcal{H}(\text{TPR}_j) + (1 - b_{t,i})\mathcal{H}(\text{FPR}_j) \right) \tag{5}$$

where $\mathcal{H}(p) = -p\log(p) - (1-p)\log(1-p)$ is entropy.[1] Crucially, the utility of a pairing is a function of both the detector's intrinsic properties $(\text{TPR}_j, \text{FPR}_j)$ and the current belief $b_{t,i}$; thus the most informative action changes dynamically as beliefs evolve. We seek a policy that minimizes the number of rounds until a target confidence is reached (e.g., all posteriors $\geq 1 - \delta$) or, equivalently, until total belief entropy falls below a threshold $\varepsilon$.

## 2.3 COMPUTATIONAL CHALLENGE

Finding the optimal item–detector assignment requires solving a maximum-weight bipartite matching problem where edge weights are information gains. While the Hungarian algorithm (Kuhn, 1955) solves this exactly, its $O(\min(N, M)^3)$ complexity per round becomes prohibitive for practical applications. This computational burden motivates our key contribution: can we design a simpler heuristic that achieves near-optimal performance? By exploiting the structure of information gain, its dependence on item uncertainty and detector diagnosticity, we develop an $O(N \log N)$ greedy algorithm that we prove performs optimally under broad conditions.

---

[1]Unless otherwise noted, all logarithms are base 2.

# 3 THE GOLD PANNING ALGORITHM

The core idea of the GOLD PANNING algorithm is to maximize information gain at each step by strategically matching the most uncertain items to the most diagnostic detectors. At each round, the algorithm ranks items by the entropy of their current belief state, prioritizing those for which we have the least certainty. Concurrently, it ranks all detectors by their intrinsic diagnosticity ($d_{\text{diag}}(j)$). The heuristic then greedily assigns the most uncertain item to the most diagnostic detector, the second-most uncertain to the second-most diagnostic, and so on.

This greedy strategy serves as an efficient proxy for solving the maximum-weight bipartite matching problem described in Section 2. While the information gain for any specific item-detector pairing, $I(Z_i; O_{ij})$, is a complex function of both the current belief $b_{t,i}$ and the detector's properties $(\text{TPR}_j, \text{FPR}_j)$, its value is maximized when item uncertainty is high ($b_{t,i} \approx 0.5$) and detector diagnosticity is high (large $|\Delta_j|$). Our heuristic directly optimizes for these conditions, sidestepping the need to compute all $N^2$ potential information gain values and solve the assignment problem.

This approach is analogous to its namesake, gold panning. An efficient prospector with several different sieves would not waste their finest, most reliable sieve on a pile of sediment they are already confident is barren. Instead, they would use their best tool on the most promising but un-inspected pile of earth to learn the most from their effort. Similarly, the GOLD PANNING algorithm applies the most discriminative tests (diagnostic positions) to the most ambiguous items (uncertain documents), ensuring that each query to the language model is maximally informative.

---

**Algorithm 1** GOLD PANNING

1: **Input:** Set of $N$ items $\{1, \ldots, N\}$; Set of $N$ detectors with parameters $\{(\text{TPR}_j, \text{FPR}_j)\}_{j=1}^{N}$; Number of rounds $T$.
2: **Initialize:** Beliefs $b_{0,i} \leftarrow 0.5$ for all $i = 1, \ldots, N$.
3: Compute $d_{\text{diag}}(j) = |\text{TPR}_j - \text{FPR}_j|$ for each detector $j$.
4: Let `sorted_detectors` be the indices of detectors sorted by $d_{\text{diag}}(j)$ in descending order.
5: **for** $t = 1$ **to** $T$ **do**
6:     Compute uncertainty $u_{t,i} \leftarrow \mathcal{H}(b_{t-1,i})$ for each item $i$.
7:     Let `sorted_items` be the indices of items sorted by $u_{t,i}$ in descending order.
8:     Define permutation $\sigma_t$ such that $\sigma_t(\texttt{sorted\_items}[k]) = \texttt{sorted\_detectors}[k]$ for $k = 1, \ldots, N$.
9:     Observe outcomes $\mathbf{O}_t = \{O_{i,\sigma_t(i)}\}_{i=1}^{N}$.
10:     **for** each item $i = 1, \ldots, N$ **do**
11:         $j \leftarrow \sigma_t(i)$.
12:         Update belief $b_{t,i}$ from $b_{t-1,i}$ using the observed outcome $O_{ij}$ via Eq. 2.
13:     **end for**
14: **end for**
15: **Output:** Final beliefs $\mathbf{b}_T = (b_{T,1}, \ldots, b_{T,N})$.

---

**Computational complexity.** The GOLD PANNING algorithm offers a significant computational advantage over solving the optimal assignment problem at each round. The primary computational cost within each iteration of the main loop is sorting the $N$ items by their uncertainty, which has a time complexity of $O(N \log N)$. All other steps within the loop, calculating uncertainties, forming the greedy assignment, and performing the Bayesian belief updates for each item, are linear operations, requiring $O(N)$ time. The initial, one-time sorting of detectors by diagnosticity costs $O(N \log N)$ and does not affect the per-round complexity.

Therefore, the overall complexity for each round of GOLD PANNING is dominated by the sorting step, resulting in an efficient $O(N \log N)$ procedure. This stands in stark contrast to the $O(N^3)$ complexity required to find the optimal assignment using the Hungarian algorithm. This substantial reduction in computational cost makes the GOLD PANNING algorithm a practical and scalable solution for real-world applications involving a large number of documents, where re-computing an optimal matching at every step would be prohibitively expensive.

## 4 THEORETICAL ANALYSIS

Our theoretical analysis aims to justify the greedy, information-theoretic heuristic at the core of GOLD PANNING. Here we present theorems establishing that the adaptive, information-maximizing nature of GOLD PANNING is designed to achieve this convergence at a faster rate than non-adaptive or random strategies. Basic properties, such as overall convergence are available in §E.

Theorems E.1 and E.2 establish that our belief state will eventually converge to the ground truth, provided the permutation strategy ensures every document is periodically assessed by an informative detector. This is a crucial foundation, but it applies equally to a random strategy as to a deliberate one. The central claim of our work, however, is not merely that convergence occurs, but that GOLD PANNING accelerates it. We now analyze the rate of convergence, proving that the greedy, information-seeking approach of GOLD PANNING maximizes the one-step reduction in uncertainty compared to less strategic methods.

**Theorem 4.1** (Greater One-Step Entropy Reduction than Random Strategy). *The* GOLD PANNING *strategy provides a greater or equal expected one-step reduction in total entropy than a random permutation strategy (e.g., PSC).*

Theorem 4.1 provides a strong justification for our greedy heuristic. However, we can establish the even stronger result, that our greedy strategy is not merely a good heuristic but is truly myopically optimal, under a mild and interpretable assumption about the nature of the detectors.

**Theorem 4.2** (Myopic Optimality for Symmetric Detectors). *Assume the detectors are symmetric, meaning $TPR_j = 1 - FPR_j$ for all positions $j$, implying that a detector is equally as good at confirming a relevant document ($TPR_j$) as it is at rejecting an irrelevant one ($1 - FPR_j$). Let documents be ordered by decreasing uncertainty (entropy) and positions be ordered by decreasing informativeness, defined as $|TPR_j - 0.5|$. Then the information gain matrix $W_{ij} = IG(b_{(i)}, p_{(j)})$ is anti-Monge, and thus the* GOLD PANNING *greedy strategy solves the maximum weight bipartite matching problem exactly and finds the optimal one-step permutation $\sigma_t^\star$ in $O(N \log N)$ time.*

The anti-Monge property means that for any four entries in the gain matrix $W$ corresponding to rows $i < k$ and columns $j < \ell$, we have $W_{ij} + W_{k\ell} \geq W_{i\ell} + W_{kj}$. Intuitively, this formalizes the idea that the "advantage" of using a more diagnostic detector (moving from column $\ell$ to $j$) is greater for a more uncertain document (moving from row $k$ to $i$). Because this "rich get richer" dynamic holds, a greedy assignment is guaranteed to be optimal. This theorem proves that for a natural class of detectors, the computationally efficient GOLD PANNING strategy is not an approximation but is identical to the optimal, but far more expensive, solution from equation 4. A proof is provided in Appendix G.

## 5 EMPIRICAL RESULTS

The primary objective of this simulation is to validate our core theoretical claims in a controlled environment. We aim to isolate and measure the performance of our GOLD PANNING (GP) algorithm against two key benchmarks: the information-agnostic baseline represented by Permutation Self-Consistency (PSC) and the computationally expensive, optimal solution provided by the Hungarian method. Our goal is to demonstrate that GP's greedy heuristic provides a substantial performance gain over the baseline and serves as an efficient, near-perfect approximation of the optimal strategy.

### 5.1 CONTROLLED SIMULATIONS

**Experimental setup.** Our simulation environment is designed to test our hypothesis across problems of varying scale. Each simulation run involves $N \in \{10, 30, 50\}$ documents and corresponding context positions. Each position $j$ is assigned a random diagnostic profile $(TPR_j, FPR_j)$ drawn from a uniform distribution, ensuring a heterogeneous environment with imperfect signals. For each run, we establish a ground truth by randomly selecting $k$ documents as relevant, where $1 \leq k \leq \sqrt{N}$ to reflect the sparsity of relevant information in real-world tasks.

All methods begin with an uninformative prior (a 0.5 belief of relevance for each document) and proceed for 20 iterations. We compare three distinct permutation strategies. **Hungarian Method:**

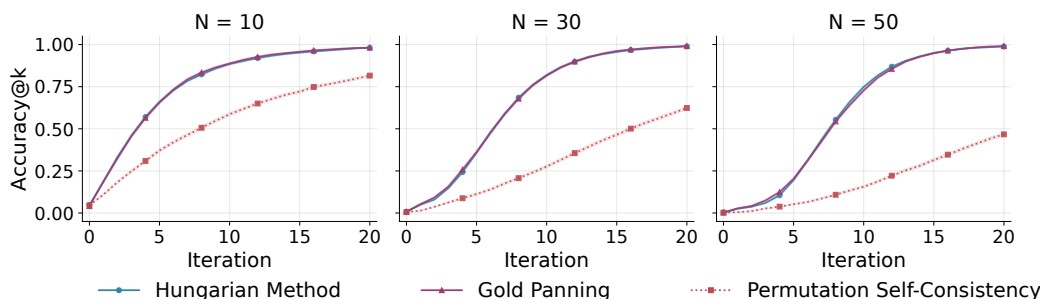

Figure 2: Performance of GOLD PANNING (GP), Hungarian Method, and Permutation Self-Consistency (PSC) baseline across 20 queries for varying numbers of documents ($N = 10, 30,$ and $50$). Accuracy@k is averaged over $10,000$ Monte Carlo runs. The results show that **GP's performance is nearly indistinguishable from the optimal Hungarian method** and across all scales **significantly outperforms the PSC baseline**.

This serves as our theoretical upper bound. At each step, it calculates the expected information gain for all $N^2$ document-position pairings and solves the assignment problem optimally. **Gold Panning (GP):** This is our proposed $O(N \log N)$ greedy heuristic. It pairs documents with the highest uncertainty (entropy) with the positions having the highest diagnosticity. **Permutation Self-Consistency (PSC):** This is our baseline, representing an information-agnostic approach. It applies a purely random permutation of documents to positions at each iteration and updates beliefs accordingly.

Performance is evaluated using **Accuracy@k**, a binary metric that equals 1 if the top-$k$ documents ranked by belief perfectly match the set of $k$ ground-truth relevant documents, and 0 otherwise. This strict criterion provides a clear signal of a method's efficiency in converging to the correct set of documents. Results are averaged over $10,000$ Monte Carlo runs to ensure statistical significance.

**Results.** The simulation results, summarized in Figure 2, strongly validate our approach and demonstrate three key findings. **First, GOLD PANNING achieves near-optimal performance.** Across all problem sizes ($N = 10, 30, 50$), the performance curve of our greedy GP strategy is nearly identical to that of the computationally intractable Hungarian method. This empirically confirms that our efficient $O(N \log N)$ heuristic is an exceptionally effective approximation of the optimal $O(N^3)$ strategy. **Second, strategic permutation is decisively superior to the random baseline.** There is a large and persistent gap between the information-seeking methods (GP and Hungarian) and the PSC baseline. For instance, with $N = 50$, GP achieves over 75% accuracy in just 10 iterations, a level PSC fails to reach even after 20 iterations. This underscores the significant efficiency gains achieved by a strategic, belief-guided assignment policy. **Finally, the advantage of GOLD PANNING scales with problem complexity.** The performance gap between GP and PSC widens as $N$ increases, highlighting the practical value of our targeted strategy, as its benefits become even more pronounced in larger, more realistic scenarios where intelligently managing uncertainty is most critical.

## 5.2 REAL-WORLD VALIDATION

**Experimental setup.** We validate GOLD PANNING on a fact-finding task where models must identify relevant information from multi-document contexts. Using the MonoRel dataset (Levy et al., 2024), we construct contexts containing one "gold" fact that answers a given question plus $N - 1$ distractor facts. The model must both answer the question and cite the supporting fact, providing a direct measure of whether each position successfully "detected" the relevant content.

We compare four strategies: (1) **Single-Shot** inference with the gold fact at a fixed position, (2) **Permutation Self-Consistency (PSC)** which randomly shuffles documents across iterations and aggregates results, (3) **GOLD PANNING (GP)** which uses our greedy algorithm to strategically reorder documents based on evolving beliefs, and (4) **Thompson Sampling (TS)** (Russo et al., 2018), which applies the Hungarian method at each iteration to identify the optimal assignment, but uses an exploration-exploitation trade-off to dynamically learn the TPR/FPR values, rather than use the calibration-computed values.

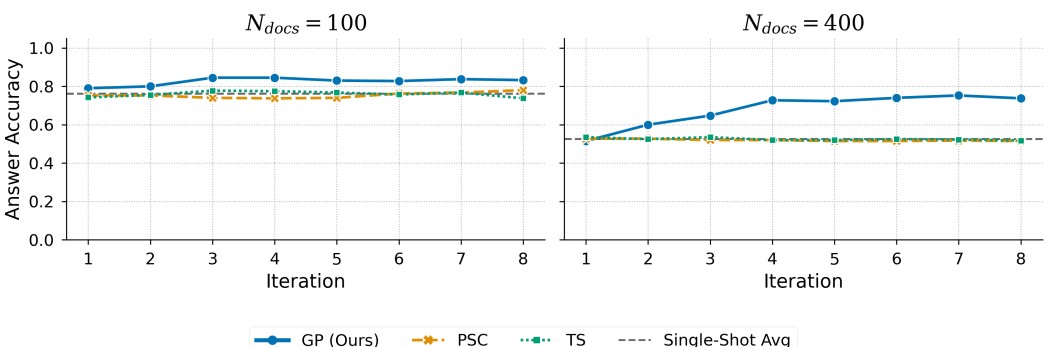

Figure 3: Performance comparison of GOLD PANNING versus baselines on GPT-4o-mini across two context sizes. The plots show answer accuracy over successive iterations. With 100 facts (left), both the PSC and TS methods largely fail to improve performance, while GP provides modest gains. With 400 facts (right), both PSC and TS continue to provide little improvement, while GP provides a roughly 34% increase (from 0.57 to 0.75), beating out both baselines. The single-shot average (TS) represents expected performance from a single query at a random position.

For each strategy, we measure the answer accuracy at subsequent iterations. We test across two context sizes (100 and 400 facts), with diagnostic parameters calibrated as described in Appendix D. Each experiment uses 100 unique samples, with the gold fact initially placed at varying positions (0%, 33%, 66%, 100% of context length) to ensure robustness across different starting conditions.

We evaluated multiple language models to understand the generality of position bias patterns. Initial analysis with GPT-5 and GPT-4o revealed insufficient position bias to facilitate a reasonable evaluation. Experiments with Gemma-3-12B-it and Gemma-27B-it revealed poor instruction following, with both models frequently struggling to return an answer, or consistently returning a citation to the final document regardless of setting. Qwen-3-7B exhibited unstable behavior, producing inconsistent outputs even with recommended settings, making reliable calibration infeasible. For systematic evaluation, we focus on GPT-4o-mini, which exhibited good instruction following and enough position bias to create heterogeneous detectors.

**Results.** Our real-world experiments demonstrate that GOLD PANNING successfully exploits position bias to improve performance where traditional approaches fail to make progress. On GPT-4o-mini with 100-fact contexts, both Permutation Self-Consistency (PSC) and the Thompson Sampling baseline (TS) show minimal improvement across iterations, essentially stagnating at the single-shot performance level. In contrast, GOLD PANNING achieves modest but consistent gains, validating our approach even in relatively homogeneous bias environments. This advantage becomes substantially more pronounced with larger contexts. At 400 facts, where position bias exhibits the characteristic U-shaped pattern shown in Figure 6, GOLD PANNING delivers a 34% performance increase (from 0.57 to 0.75 accuracy) while both baseline methods again fail to improve meaningfully beyond their initial performance. The inability of PSC to improve despite multiple permutations particularly highlights the value of strategic reordering over random shuffling. While PSC treats each permutation independently and learns nothing from previous iterations, GOLD PANNING leverages the accumulated belief state to make increasingly informed assignment decisions. These results confirm our theoretical predictions that systematic position bias, when properly characterized and exploited through our greedy assignment algorithm, transforms from a hindrance into a powerful signal for efficient information discovery in multi-document contexts.

## 6 RELATED WORK

**Exploiting systematic biases for efficiency.** While systematic biases are typically viewed as obstacles, prior work has shown they can be valuable signals. In human–computer interaction, cognitive biases have been leveraged for interface optimization (Gajos & Weld, 2005; Gajos et al., 2008),

to improve recommender systems (Tversky & Kahneman, 1981; Cialdini & Cialdini, 2007; Lerman & Hogg, 2010; 2014), and guide (for better or worse) user behavior (Fogg, 2002; Gray et al., 2018; Mathur et al., 2019; Di Geronimo et al., 2020). Similarly, in active learning, predictable failure points of models have been exploited to guide query selection and accelerate learning (Settles, 2009; Zhang et al., 2022). Our work extends this philosophy to language models, showing that position bias, when understood, provides a rich signal for efficient information discovery.

**Position bias in LLMs.** Position bias in LLMs has been studied extensively (Wang et al., 2023b; Zheng et al., 2023; Liu et al., 2024, inter alia.). While recent work proposes mitigation strategies, such as context compression (Jiang et al., 2024), attention manipulation (Hsieh et al., 2024; Wang et al., 2024), and specialized fine-tuning (Xiong et al., 2024), these methods largely treat position bias as noise to remove. As these mitigation can be computationally costly and complete elimination may be infeasible, we take an orthogonal approach: consistent biases, though problematic for direct inference, constitute reliable signals that can be exploited for efficient information discovery. Work closest to ours, Permutation Self-Consistency (PSC) (Tang et al., 2024), also uses multiple document orderings but still treats position bias as noise to be averaged away through random permutations.

**Bandit algorithms and information acquisition.** Decision-making under uncertainty has long been studied via multi-armed bandits (MABs) (Slivkins, 2019; Wang et al., 2005; Gai et al., 2010; 2012; Chen et al., 2013). Classical formulations typically assume that the reward of an action depends only on its inherent (unknown) utility. Our GOLD PANNING BANDITS framework instead models *state-coupled* utility that depends jointly on *what* is tested (the item's belief state) and *how* it is tested (the detector's diagnostic properties). Recent work has explored using MAB frameworks to optimize LLM performance, with Duan et al. (2025) modeling context chunks within a bandit setting to generate high-quality responses. While also a bandit-based selection problem, their approach is a *training-time strategy* designed to improve performance via preference optimization. In direct contrast, our GOLD PANNING algorithm is an *inference-time strategy* that works with any existing LLM to reduce the cost of discovering relevant information, *requiring no model retraining.*

## 7 CONCLUSION

Our proposed algorithm, GOLD PANNING, implements an efficient greedy strategy that matches the most uncertain documents with the most informative context positions. Both theoretical analysis and empirical validation confirm that this computationally efficient approach achieves a nearly optimal rate of uncertainty reduction. In practical terms, GOLD PANNING can rapidly identify relevant document, offering a significant reduction in computational cost without requiring any model retraining. By turning a systematic model bias from a liability into an asset, our work provides a powerful new tool for efficient, inference-time optimization in knowledge-intensive NLP tasks.

## ETHICS STATEMENT

This work presents a method for improving the computational efficiency of information retrieval from large language models, which has positive environmental implications through reduced energy consumption. Our approach leverages existing model biases rather than introducing new biases or harmful behaviors. All experiments were conducted using publicly available datasets and pre-trained models, with no involvement of human subjects or collection of personal data. We acknowledge that improved efficiency in LLM querying could enable broader deployment of these systems; however, we believe the environmental and accessibility benefits outweigh potential risks. Gemin-2.5-Pro was used to check for spelling and grammatical mistakes, as well as to aid in clarifying some sentences.

## REPRODUCIBILITY STATEMENT

To ensure reproducibility of our results, we provide comprehensive implementation details throughout the paper and appendices. Algorithm 1 presents the complete GOLD PANNING procedure with explicit pseudocode. §5.1 details our simulation environment, including the exact distributions used for generating detector parameters and the Monte Carlo procedure. §D provides instructions for calibrating TPR and FPR parameters.

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

## A    INFORMATION GAIN

**Proposition A.1.** *The information gain $\mathcal{I}(Z_i; O_{ij} \mid b_{t,i})$ from testing an item $i$ with current belief $b_{t,i}$ using a detector $j$ with parameters $(\mathrm{TPR}_j, \mathrm{FPR}_j)$ is given by:*

$$\mathcal{I}(Z_i; O_{ij} \mid b_{t,i}) = \mathcal{H}(\mathrm{FPR}_j + b_{t,i}\Delta_j) - \Big(b_{t,i}\mathcal{H}(\mathrm{TPR}_j) + (1 - b_{t,i})\mathcal{H}(\mathrm{FPR}_j)\Big), \qquad (6)$$

*where $\Delta_j = \mathrm{TPR}_j - \mathrm{FPR}_j$ and $\mathcal{H}(\cdot)$ is the binary entropy function.*

*Proof.* We use the definition of mutual information, which is $\mathcal{I}(Z_i; O_{ij}) = \mathcal{H}(O_{ij}) - \mathcal{H}(O_{ij} \mid Z_i)$. All entropies are calculated with respect to the current belief, $b_{t,i}$.

**Step 1: Calculate the marginal entropy $\mathcal{H}(O_{ij})$.** This is the entropy of the observation $O_{ij}$ before the true state $\mathcal{H}_i$ is known. The probability of observing $O_{ij} = 1$ is the marginal probability, averaged over the possible states of $Z_i$:

$$\begin{aligned}
\Pr(O_{ij} = 1) &= \Pr(O_{ij} = 1 \mid Z_i = 1)\Pr(Z_i = 1) + \Pr(O_{ij} = 1 \mid Z_i = 0)\Pr(Z_i = 0) \\
&= (\mathrm{TPR}_j \cdot b_{t,i}) + (\mathrm{FPR}_j \cdot (1 - b_{t,i})) \\
&= \mathrm{FPR}_j + b_{t,i}(\mathrm{TPR}_j - \mathrm{FPR}_j) \\
&= \mathrm{FPR}_j + b_{t,i}\Delta_j
\end{aligned}$$

Since the observation $O_{ij}$ is a Bernoulli random variable with this probability, its entropy is $\mathcal{H}(O_{ij}) = \mathcal{H}(\mathrm{FPR}_j + b_{t,i}\Delta_j)$.

**Step 2: Calculate the conditional entropy $\mathcal{H}(O_{ij} \mid H_i)$.** This is the expected entropy of the observation, where the expectation is taken over the true state $Z_i$:

$$\begin{aligned}
\mathcal{H}(O_{ij} \mid Z_i) &= \Pr(Z_i = 1) \cdot \mathcal{H}(O_{ij} \mid Z_i = 1) + \Pr(Z_i = 0) \cdot \mathcal{H}(O_{ij} \mid Z_i = 0) \\
&= b_{t,i} \cdot \mathcal{H}(\mathrm{TPR}_j) + (1 - b_{t,i}) \cdot \mathcal{H}(\mathrm{FPR}_j)
\end{aligned}$$

This follows because when $Z_i = 1$, the observation $O_{ij}$ is a Bernoulli variable with parameter $\mathrm{TPR}_j$, and when $Z_i = 0$, it is a Bernoulli variable with parameter $\mathrm{FPR}_j$.

**Step 3: Combine terms.** Substituting the results from Step 1 and Step 2 into the definition of mutual information yields the final expression:

$$\mathcal{I}(Z_i; O_{ij} \mid b_{t,i}) = \mathcal{H}(\mathrm{FPR}_j + b_{t,i}\Delta_j) - \Big(b_{t,i}\mathcal{H}(\mathrm{TPR}_j) + (1 - b_{t,i})\mathcal{H}(\mathrm{FPR}_j)\Big)$$

$\square$

## B    ROBUSTNESS TO NOISE IN BIAS ESTIMATION

A core assumption of the GOLD PANNING heuristic is that we have access to accurate estimates of each detector's parameters, namely its True Positive Rate ($TPR_j$) and False Positive Rate ($FPR_j$). In practical applications, these must be estimated from calibration data and are therefore subject to noise. To assess the robustness of our proposed strategy to such estimation errors, we conducted a simulation experiment to quantify its performance under increasing levels of uncertainty.

### B.1    SIMULATION SETUP

We augment the simulation described in §5.1 by introducing Gaussian noise to the detector parameters that are available to the agent. At each iteration $t$, for each detector $j$, the agent does not use the true parameters $(TPR_j, FPR_j)$ to make its assignment decision. Instead, it uses noisy estimates:

$$\widehat{TPR}_{j,t} = \mathrm{clamp}(TPR_j + \epsilon_1, 0, 1)$$
$$\widehat{FPR}_{j,t} = \mathrm{clamp}(FPR_j + \epsilon_2, 0, 1)$$

where $\epsilon_1, \epsilon_2 \sim \mathcal{N}(0, \sigma_{\text{noise}}^2)$. Crucially, the observations themselves are still generated from the *true* detector parameters; the noise only corrupts the agent's perception of these parameters, directly affecting its assignment policy. We evaluated the performance of the GOLD PANNING strategy across four distinct noise levels, defined by the standard deviation $\sigma_{\text{noise}}$ of the injected noise. These levels were chosen to represent a range from perfect knowledge to significant uncertainty:

- **Perfect:** $\sigma_{\text{noise}} = 0.0$. This is the baseline scenario from the main paper.

- **Low Noise:** $\sigma_{\text{noise}} = 0.0051$. This level is calibrated such that there is a 95% probability that the agent's parameter estimate is within $\pm 0.01$ of the true value.

- **Medium Noise:** $\sigma_{\text{noise}} = 0.0255$, calibrated for a 95% probability of being within $\pm 0.05$ of the true value.

- **High Noise:** $\sigma_{\text{noise}} = 0.0510$, calibrated for a 95% probability of being within $\pm 0.10$ of the true value.

## B.2 RESULTS

The results, depicted in Figure 4, demonstrate that the GOLD PANNING strategy exhibits substantial robustness and degrades gracefully as noise increases.

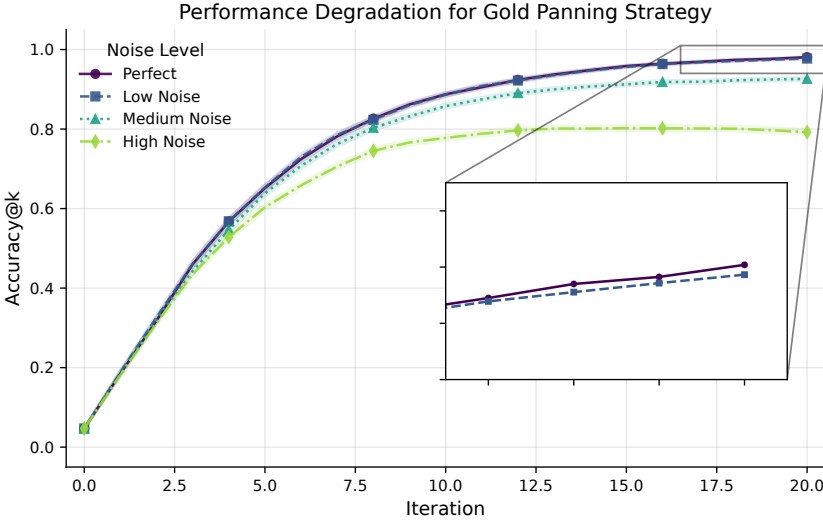

Figure 4: **Performance Degradation of the GOLD PANNING Strategy under Noisy Parameter Estimates.** The plot shows performance over 20 iterations, averaged across $10,000$ Monte Carlo runs, for four levels of noise in the agent's estimates of detector TPR and FPR. The inset provides a magnified view of the final iterations, highlighting the subtle divergence between the "Perfect" and "Low Noise" scenarios. The strategy shows strong resilience to minor estimation errors and degrades gracefully under more significant noise.

Under the "Low Noise" condition, performance is nearly indistinguishable from the "Perfect" knowledge scenario, as highlighted by the inset. This indicates that the heuristic is highly resilient to minor inaccuracies in parameter estimation, a crucial property for real-world deployment. As the noise level increases to "Medium" and "High," we observe a clear but bounded decrease in terminal accuracy. The strategy's ability to quickly improve accuracy in the initial iterations remains largely intact, but the final convergence point is lower. This graceful degradation, rather than a catastrophic failure, confirms that the core principle of matching uncertain items to diagnostic detectors remains effective even when the measures of uncertainty and diagnosticity are imperfect. The results strongly suggest that GOLD PANNING is a practical and robust heuristic for real-world applications where detector parameters cannot be known with perfect precision.

## C EFFECT OF ENVIRONMENT HOMOGENEITY

A core premise of the GOLD PANNING heuristic is that there is a strategic advantage to be gained by intelligently matching items to detectors. This implicitly assumes that the detectors are meaningfully different from one another, that is, that the environment is **heterogeneous**. In this appendix, we explore the boundary conditions of this assumption by evaluating how the performance of GOLD

PANNING changes as the detector environment shifts from highly heterogeneous to nearly homogeneous.

## C.1 SIMULATION SETUP

We augment the simulation described in §5.1 to generate detector parameters from a symmetric Beta distribution, $Beta(\alpha, \alpha)$, to precisely control the level of environmental homogeneity. The **concentration parameter**, $\alpha$, governs the variance of the detector parameters ($TPR_j, FPR_j$):

- **Low Concentration ($\alpha < 1$):** The Beta distribution is U-shaped, producing detector parameters clustered near 0 and 1. This results in a highly *heterogeneous* environment, with a wide spread of detector diagnosticities.
- **High Concentration ($\alpha \gg 1$):** The Beta distribution is sharply peaked at 0.5. This means all detectors have very similar $TPR_j$ and $FPR_j$ values, resulting in a *homogeneous* environment where there is little to no difference in diagnosticity across detectors.

We ran $10,000$ Monte Carlo simulations for each of 20 concentration levels, logarithmically spaced from $10^{-1}$ to $10^{2}$.

## C.2 RESULTS

The results, presented in Figure 5, demonstrate that the strategic advantage of GOLD PANNING is fundamentally linked to the heterogeneity of the detector environment.

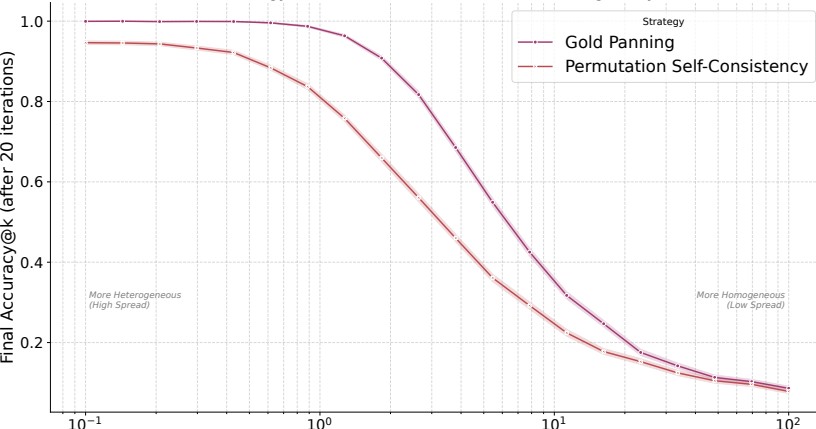

Figure 5: Performance of GOLD PANNING vs. PERMUTATION SELF-CONSISTENCY (PSC) as a function of environment homogeneity. The x-axis plots the Beta concentration parameter ($\alpha$) on a log scale, where higher values correspond to more homogeneous detectors. The y-axis shows the final ranking accuracy after 20 iterations. The advantage of GOLD PANNING is largest in heterogeneous environments (low concentration) and vanishes as the environment becomes homogeneous.

In a **heterogeneous environment** (low concentration, left side of the plot), GOLD PANNING significantly outperforms PERMUTATION SELF-CONSISTENCY (PSC). This is because a wide variance in detector quality ($|TPR_j - FPR_j|$) creates an opportunity for strategic assignment. GOLD PANNING successfully exploits this by pairing the most uncertain items with the most diagnostic detectors, accelerating uncertainty reduction and leading to higher accuracy.

Conversely, as the environment becomes more **homogeneous** (high concentration, right side of the plot), the performance of the two strategies converges. When all detectors have nearly identical properties, there is no longer a "best" detector to assign to an uncertain item. The core logic of the greedy heuristic is nullified, as any permutation yields roughly the same expected information gain. In this scenario, the strategic assignment of GOLD PANNING provides no more benefit than a random one, and its performance becomes indistinguishable from that of PSC.

This analysis confirms that the value of the GOLD PANNING framework lies in its ability to effectively exploit environmental heterogeneity. The algorithm is most impactful in scenarios where detector quality varies, which is a common characteristic of real-world systems like the positional biases found in large language models.

## D  ESTIMATING $\mathrm{TPR}_j$ AND $\mathrm{FPR}_j$

### D.1  CALIBRATION EXPERIMENTAL DESIGN

To apply the GOLD PANNING algorithm to real language models, we need to estimate the diagnostic parameters $(TPR_j, FPR_j)$ for each context position $j$. We accomplish this through a controlled calibration experiment that measures how reliably each position surfaces relevant information. Our calibration procedure uses a fact-checking task where the model must identify which fact in a multi-document context answers a given question. For each trial, we construct a context containing exactly one "gold" fact (the correct answer) and $N-1$ distractor facts, then ask the model to answer a question and cite the supporting fact. By systematically varying the gold fact's position across many trials, we can measure each position's diagnostic properties.

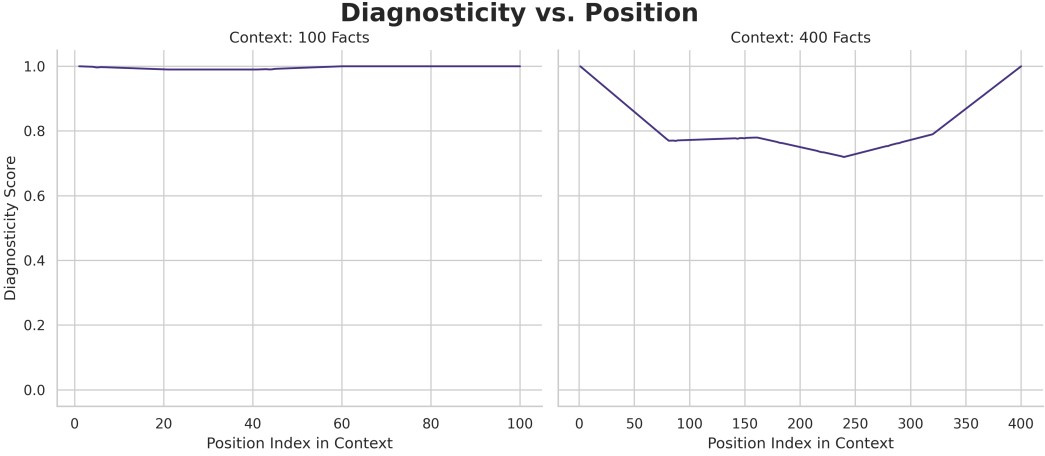

Figure 6: Diagnosticity as a function of position for GPT-4o-mini with varying context sizes. The 100-fact context exhibits nearly uniform diagnosticity across all positions, representing a homogeneous detector environment where strategic reordering provides minimal benefit. In contrast, the 400-fact context shows pronounced position bias with a characteristic U-shaped pattern, creating the heterogeneous environment where GOLD PANNING significantly outperforms random baselines.

### D.2  PARAMETER ESTIMATION

For a given context size $N$ and model, we estimate parameters as follows.

**True Positive Rate** ($TPR_j$)**:** This measures how often position $j$ successfully "detects" relevant content when it is actually present. We calculate it as:

$$TPR_j = \Pr(\text{model cites position } j \mid \text{gold fact is at position } j)$$
$$= \frac{\#\text{trials where model correctly cites } j \text{ when gold is at } j}{\#\text{trials where gold fact is placed at position } j}$$

**False Positive Rate** ($FPR_j$)**:** This measures how often position $j$ incorrectly appears salient when it contains irrelevant content. We calculate it as:

$$FPR_j = \Pr(\text{model cites position } j \mid \text{gold fact is NOT at position } j)$$
$$= \frac{\#\text{trials where model incorrectly cites } j \text{ when gold is elsewhere}}{\#\text{trials where gold fact is NOT placed at position } j}$$

### D.3 IMPLEMENTATION DETAILS

In practice, we implement this calibration using the MonoRel dataset (Levy et al., 2024), which provides fact-question pairs with verified answers. For each calibration run: (1) we sample $S$ fact-question pairs from the dataset (typically S = 100-500 for reliable estimates). (2) For each sample, we create a context with the gold fact placed at various positions (i.e. at 0%, 20%, 40%, 60%, 80%, 100% of the total facts). (3) Fill remaining positions with randomly selected distractor facts, filtered to ensure none are the gold fact. (4) Query the model for both an answer and citation index, and record whether the citation matches the gold position.

From the resulting data, we compute the $TPR$, $FPR$, and diagnosticities for each position, as illustrated in Figure 6.

## E CONVERGENCE GUARANTEE

Here we establish the foundational results that the iterative Bayesian belief updating process is guaranteed to converge regardless of the strategy employed.

**Theorem E.1** (Belief Entropy Converges for Any Strategy). *The total entropy $\{\sum_i \mathcal{H}(b_{t,i})\}_{t \geq 0}$ is a non-negative supermartingale and converges almost surely to a finite limit $\mathcal{H}_\infty \geq 0$.*

*Proof.* By the nonnegativity of conditional mutual information (Cover & Thomas, 2005, Corollary to Theorem 2.6.3),
$$\mathbb{E}[\mathcal{H}(\mathbf{b_{t+1}})] \leq \mathcal{H}(\mathbf{b_t})$$
, and thus $\mathcal{H}(\mathbf{b_t})$ is a supermartingale. By the Martingale Convergence Theorem (Durrett, 2019, Theorem 5.2.12),
$$\mathcal{H}(\mathbf{b_t}) \xrightarrow[t \to \infty]{a.s.} \mathcal{H}_\infty$$
□

Theorem E.1 guarantees that our uncertainty about document relevance will eventually stop decreasing, regardless of the strategy used. While this ensures stability, it does not guarantee that the beliefs converge to the correct values. For example, the entropy could stabilize at a non-zero value if a document is never placed in an informative position. Theorem E.2 provides the crucial next step, establishing the conditions under which our beliefs are guaranteed to converge to the ground truth, a property known as posterior consistency (Diaconis & Freedman, 1986).

**Theorem E.2** (Posterior Consistency Under Minimal Informativity). *Assume:*

*(i) (Minimal informativity) There exists at least one position $j$ with diagnosticity $d_{\text{diag}}(j) > 0$.*

*(ii) (Persistent exposure) For each document $i$, the policy assigns $i$ to some informative position (i.e., a position $j$ with $d_{\text{diag}}(j) > 0$) infinitely often almost surely.*

*(iii) (Standard regularity) Conditional on $Z_i$, observations from a fixed position are independent across time with stationary parameters $TPR_j$ and $FPR_j$.*

*Then for every document $i$, the posterior belief $b_{t,i}$ converges almost surely to the true relevance state $Z_i$ as $t \to \infty$:*
$$b_{t,i} \xrightarrow[t \to \infty]{a.s.} Z_i.$$
*Consequently, the limiting total entropy is $\mathcal{H}_\infty = 0$ almost surely.*

*Proof.* Fix $d_i$ and define the posterior log-odds $l_{t,i} = \log\left(\frac{b_{t,i}}{1 - b_{t,i}}\right)$. Let $j_t$ be the position used at time $t$. Bayes' rule gives,
$$b_{t+1,i} = \Pr(Z_i = 1 | O_t, j_t = j) = \frac{b_{t,i} \Pr(O_t | Z_i = 1)}{b_{t,i} \Pr(O_t | Z_i = 1) + (1 - b_{t,i}) \Pr(O_t | Z_i = 0)}.$$
Hence
$$\frac{b_{t+1,i}}{1 - b_{t+1,i}} = \frac{b_{t,i}}{1 - b_{t,i}} \cdot \frac{\Pr(O_t | Z_i = 1)}{\Pr(O_t | Z_i = 0)}, \quad l_{t+1,i} = l_{t,i} + \epsilon_{t,j},$$

where

$$\epsilon_{t,j} = \log\Big(\frac{\Pr(O_t \mid Z_i = 1)}{\Pr(O_t \mid Z_i = 0)}\Big) = \mathbf{1}_{O_t=1} \cdot \log\Big(\frac{TPR_j}{FPR_j}\Big) + \mathbf{1}_{O_t=0} \cdot \log\Big(\frac{1 - TPR_j}{1 - FPR_j}\Big).$$

For a fixed informative position $j$ with $d_{\mathrm{diag}}(j) > 0$ (guaranteed to exist form assumption (i)), assumption (iii) makes $\epsilon_{t,j}$ i.i.d., with

$$\mathbb{E}[\epsilon_{t,j}|Z_i = 1] = D_{KL}(\mathrm{Bernoulli}(TPR_j)||\mathrm{Bernoulli}(FPR_j)) > 0$$
$$\mathbb{E}[\epsilon_{t,j}|Z_i = 0] = -D_{KL}(\mathrm{Bernoulli}(FPR_j)||\mathrm{Bernoulli}(TPR_j)) < 0$$

By (ii), the document hits such a $j$ infinitely often a.s., and the SLLN (Durrett, 2019, Theorem 2.4.1) on that sequence implies that the partial sums of $\epsilon_{t,j}$ diverge to $+\infty$ if $Z_i = 1$ and $-\infty$ if $Z_i = 0$. Therefore, $l_{t,i} \to +\infty$ (resp. $-\infty$), so $b_{t,i} \to 1$ (resp. 0) almost surely. Thus

$$\mathcal{H}(b_t) \xrightarrow[t\to\infty]{a.s.} 0.$$

$\square$

## F    PROOF OF THEOREM 4.1

**Theorem F.1** (Greater One-Step Entropy Reduction than Random Strategy). *The* GOLD PANNING *strategy provides a greater or equal expected one-step reduction in total entropy than a random permutation strategy (e.g., PSC).*

*Proof.* Let the ordered document uncertainties be $u_{(1)} \geq u_{(2)} \geq \cdots \geq u_{(N)}$ and the ordered position diagnosticities be $d_{(1)} \geq d_{(2)} \geq \cdots \geq d_{(N)}$. The information gain from pairing the document with uncertainty $u_i$ with the position of diagnosticity $d_j$ is $\mathcal{I}_{ij} = f(u_i, d_j)$. As shown in §A, this function is monotonically increasing in both its arguments. The total information gain for any permutation $\sigma$ of the positions is $\mathcal{G}(\sigma) = \sum_{i=1}^{N} f(u_{(i)}, d_{\sigma(i)})$.

The GOLD PANNING strategy, $\sigma_{GP}$, pairs elements of the same rank, yielding a total gain of $\mathcal{G}(\sigma_{GP}) = \sum_{i=1}^{N} f(u_{(i)}, d_{(i)})$.

By the **rearrangement inequality** (Hardy et al., 1934), for any two sequences ordered in the same direction, the sum of function applications is maximized when elements of the same rank are paired together. Therefore, the gain from the GOLD PANNING strategy is the maximum possible one-step gain:

$$\mathcal{G}(\sigma_{GP}) = \sum_{i=1}^{N} f(u_{(i)}, d_{(i)}) \geq \sum_{i=1}^{N} f(u_{(i)}, d_{\sigma(i)}) \quad \forall \sigma \in S_N$$

Thus, the GOLD PANNING strategy is guaranteed to reduce uncertainty at least as fast as, and typically faster than, a random permutation strategy in any single step. $\square$

## G    PROOF OF THEOREM 4.2

**Theorem G.1** (Myopic Optimality for Symmetric Detectors). *Assume the detectors are symmetric, meaning $TPR_j = 1 - FPR_j$ for all positions $j$, implying that a detector is equally as good at confirming a relevant document ($TPR_j$) as it is at rejecting an irrelevant one ($1 - FPR_j$). Let documents be ordered by decreasing uncertainty (entropy) and positions be ordered by decreasing informativeness, defined as $|TPR_j - 0.5|$. Then the information gain matrix $W_{ij} = IG(b_{(i)}, p_{(j)})$ is anti-Monge, and thus the* GOLD PANNING *greedy strategy solves the maximum weight bipartite matching problem exactly and finds the optimal one-step permutation $\sigma_t^\star$ in $O(N \log N)$ time.*

*Proof.* To prove that the greedy strategy is optimal, we must show that the information gain matrix $W$ with entries $W_{ij} = \mathcal{I}(b_{(i)}, p_{(j)})$ is anti-Monge. A matrix is anti-Monge if for any two rows $i < k$ and two columns $j < l$, it satisfies the property:

$$W_{ij} + W_{kl} \geq W_{il} + W_{kj}$$

This can be rearranged into an "increasing differences" condition:

$$W_{ij} - W_{il} \geq W_{kj} - W_{kl}$$

In our context, $i < k$ implies document $(i)$ is more uncertain than document $(k)$, so $\mathcal{H}(b_{(i)}) \geq \mathcal{H}(b_{(k)})$. Similarly, $j < l$ implies position $(j)$ is more informative than position $(l)$, so $|TPR_{(j)} - 0.5| \geq |TPR_{(l)} - 0.5|$. The inequality means that the advantage gained by using the more informative position $(j)$ instead of $(l)$ is greater for the more uncertain document $(i)$ than it is for the less uncertain document $(k)$.

We begin with the formula for information gain (equation 5):

$$\mathcal{I}(Z_i; O_{ij} \mid b_i) = \mathcal{H}(FPR_j + b_i \Delta_j) - (b_i \mathcal{H}(TPR_j) + (1 - b_i)\mathcal{H}(FPR_j))$$

Under the symmetric detector assumption, $TPR_j = 1 - FPR_j$; denote $p_j = TPR_j$. Then:

- $FPR_j = 1 - p_j$

- $\Delta_j = TPR_j - FPR_j = p_j - (1 - p_j) = 2p_j - 1$

- $\mathcal{H}(FPR_j) = \mathcal{H}(1 - p_j) = \mathcal{H}(p_j)$

Substituting these into the gain equation simplifies the second term:

$$b_i \mathcal{H}(p_j) + (1 - b_i)\mathcal{H}(p_j) = \mathcal{H}(p_j)$$

The first term becomes $\mathcal{H}((1 - p_j) + b_i(2p_j - 1))$. Thus, the information gain for a symmetric detector is:

$$\mathcal{I}(b, p) = \mathcal{H}((1 - p) + b(2p - 1)) - \mathcal{H}(p)$$

We define the "advantage" function $A(b) = \mathcal{I}(b, p_{(j)}) - \mathcal{I}(b, p_{(l)})$, where $p_{(j)}$ is more informative than $p_{(l)}$. We need to show that $A(b_{(i)}) \geq A(b_{(k)})$ whenever $\mathcal{H}(b_{(i)}) \geq \mathcal{H}(b_{(k)})$. This is equivalent to showing that $A(b)$ is maximized when uncertainty $\mathcal{H}(b)$ is maximized, which occurs at $b = 0.5$.

The advantage is:

$$A(b) = \left[ \mathcal{H}((1 - p_{(j)}) + b(2p_{(j)} - 1)) - \mathcal{H}(p_{(j)}) \right] - \left[ \mathcal{H}((1 - p_{(l)}) + b(2p_{(l)} - 1)) - \mathcal{H}(p_{(l)}) \right]$$

The derivative of the entropy function $\mathcal{H}(x)$ is $\mathcal{H}'(x) = \log_2(\frac{1-x}{x})$, and so the derivative of our advantage function is:

$$A'(b) = (2p_{(j)} - 1)\mathcal{H}'(x_j(b)) - (2p_{(l)} - 1)\mathcal{H}'(x_l(b))$$

where $x_j(b) = (1 - p_{(j)}) + b(2p_{(j)} - 1)$ and $x_l(b) = (1 - p_{(l)}) + b(2p_{(l)} - 1)$.

At maximum uncertainty ($b = 0.5$), we have $x_j(0.5) = 0.5$ and $x_l(0.5) = 0.5$. Since $\mathcal{H}'(0.5) = \log_2(1) = 0$, we get $A'(0.5) = 0$. Further analysis of the second derivative shows this is a maximum. Because the function $A(b)$ is symmetric and has a unique maximum at $b = 0.5$, its value decreases as $b$ moves away from $0.5$.

Since document uncertainty $\mathcal{H}(b)$ also decreases as $b$ moves away from $0.5$, a higher uncertainty $\mathcal{H}(b_{(i)}) \geq \mathcal{H}(b_{(k)})$ implies that $b_{(i)}$ is closer to $0.5$ than $b_{(k)}$ is. Therefore, $A(b_{(i)}) \geq A(b_{(k)})$, which satisfies the anti-Monge condition.

Because the gain matrix is anti-Monge, the maximum weight bipartite matching problem can be solved optimally with a greedy algorithm. The GOLD PANNING algorithm sorts documents by uncertainty and positions by informativeness and pairs them accordingly. This procedure is computationally dominated by sorting, resulting in an $O(N \log N)$ time complexity. $\qquad \square$

## H    LIMITATIONS

While our framework demonstrates significant promise, it is important to acknowledge its limitations, which also point to valuable directions for future work.

**Calibration Overhead:** The GOLD PANNING algorithm relies on having a pre-estimated diagnostic profile for each context position. This requires a one-time calibration phase for a given LLM to characterize its biases. Although our simulations show the method is robust to noise in these estimates, this initial setup cost may be a consideration for some use cases.

**Conditional Independence Assumption:** Our formal model assumes that, given a document's true relevance, the model's observation at one position is independent of the documents ar other positions within the same context. While our empirical results suggest position effects are dominant and the assumption is a practical abstraction, the nature of the transformer attention mechanism violates this assumption in principle. Complex inter-document relationships could present challenges.

**Stationary Bias Profile:** We assume a stationary bias profile for the LLM. However, it is possible that these biases could shift depending on the specific task, domain, or query at hand. Future work should investigate methods for dynamically updating the detector parameters online.

**Binary Relevance:** Finally, the current GOLD PANNING BANDITS framework models relevancy as a binary state (relevant or not). Many real-world scenarios involve graded or nuanced levels of relevance. Extending the belief-updating mechanism and information gain objective to accommodate a continuous or ordinal relevance scale is a non-trivial by promising avenue for future work.

