# OpenReview forum: "Gold Panning: Turning Positional Bias into Signal for Multi-Document LLM Reasoning"
_ICLR.cc/2026/Conference — ICLR 2026 Conference Withdrawn Submission_

### Official Review · Reviewer_rfxc · 2025-10-17

**Soundness:** 4
**Presentation:** 3
**Contribution:** 2
**Rating:** 4
**Confidence:** 3

**Summary:**

This paper proposes a novel method to more efficiently rank documents’ relevance. The main contribution is demonstrating that an LLM's positional bias can be transformed from a weakness into a strength.

It introduces the GOLD PANNING algorithm, which strategically reorders documents in the context. By placing the most uncertain documents in the most informative positions, it more efficiently identifies relevant information, significantly reducing the number of queries required without any model retraining.

**Strengths:**

1. The perspective is very novel. This represents the first application of bandit algorithms to long-context problems, and the first time of leveraging the position bias as a strength.
2. A rigorous theoretical analysis of the algorithm, showing the solid mathematical and algorithmic foundation of the author.
3. The problem description is clear.

**Weaknesses:**

1. The practical significance is relatively small. If you only want to determine each document's relevance with the query, why not just judge them using LLM (such as gpt-5), which should be faster, cheaper, more direct and more accurate. For example, you can input one document and the query each time and ask gpt-5 whether it's relevant; or input the document list and ask gpt-5 to rank every document's relevance.

2. Some important content is not detailed in main content, making it hard to comprehend the whole method. For example, I wonder how to transform the LLM's output into a signal to reflect relevance, but this information is in appendix instead of in the main content.

3. As mentioned by the author in section 5.2, high demands are placed on the model selection: powerful models don't have obvious position bias while weak models can't follow instructions. This limits the generalization of this method in real world applications.

**Questions:**

Have you tried to directly ask powerful LLMs such as gpt-5 to do the relevance ranking? Is it a faster, cheaper and easier way?

---

> ### Author Response · Authors · 2025-11-21
> **Rebuttal**
>
> Thank you for your review and your high praise for our work's **"rigorous theoretical analysis"** and **"solid mathematical and algorithmic foundation."** We are glad you recognized the novelty of our perspective in being the **"first application of bandit algorithms to long-context problems"** and the first to leverage position bias as a strength.
>
> Just as a reminder, here are the contributions of our work:
>
> 1. **A New Framework:** We introduce **GOLD PANNING BANDITS**, the first formalism to re-conceptualize LLM position bias as a signal for efficient information discovery, framing it as a new class of combinatorial bandits.
> 2. **An Efficient Algorithm:** We propose **GOLD PANNING**, an efficient $O(N \log N)$ ($N$ being the number of docs) greedy algorithm that strategically matches uncertain documents to informative context positions to maximize information gain per query.
> 3. **Theoretical Guarantees:** We provide theoretical analysis proving our greedy algorithm is myopically optimal and achieves a faster rate of uncertainty reduction than random strategies (random document ordering).
> 4. **Empirical Validation:** We show that GOLD PANNING identifies relevant documents using fewer queries in simulation and real-world tasks.
>
> Below are our responses to your specific comments:
>
> ---
>
> > The practical significance is relatively small... why not just judge them using LLM (such as gpt-5), which should be faster, cheaper, more direct and more accurate. ... Have you tried to directly ask powerful LLMs such as gpt-5 to do the relevance ranking?
> >
>
> Thank you for this crucial question, as it allows us to clarify the problem setup and the practical significance of our work. The scenario our paper addresses is **multi-document reasoning** (e.g., RAG, multi-document QA), where a model must synthesize an answer from a large collection of N documents provided in a single context. The alternative methods you proposed are either the very baseline we are outperforming or are dramatically less efficient.
>
> 1. **"Input one document... each time" (The** N**-Query Approach):** This approach is N **times more expensive** than a single pass of our method. For our N=400 experiment, this would require **400 separate LLM queries**. Our method (and the PSC baseline) makes one query (over all 400 documents) per iteration.
> 2. **"Input the document list and ask gpt-5 to rank" (The 1-Query Approach):** This is the **single-shot baseline**. This simple approach is known to be unreliable due to the very **position bias** our paper studies. A model like GPT-5 will not rank all 400 documents accurately; it will disproportionately favor those at the beginning and end of the context. Our own "Single-Shot Avg" baseline in Figure 3 confirms this method's poor performance.
>
> Therefore, our method provides a way to get an accurate answer from a large document set in a small handful of queries (T), which is vastly cheaper than the N-query approach and vastly more accurate than the 1-query approach.
>
> **Planned changes:** We will add a paragraph to our introduction to more clearly situate our problem setting against these N-query and 1-query alternatives to make the practical significance explicit.
>
> ---

---

> > ### Author Response · Authors · 2025-11-21
> > **Rebuttal (cont.)**
> >
> > > As mentioned by the author in section 5.2, high demands are placed on the model selection... This limits the generalization of this method in real world applications.
> > >
> >
> > This is a critical point, and we are pleased to provide a significant update in response. We have **successfully expanded our experimental setup** to include the additional models and datasets you requested.
> >
> > As we noted in our original submission, our initial challenges with models like Gemma were not due to a failure of our method, but rather due to **"poor instruction following" and "unstable behavior"** that made reliable calibration and evaluation infeasible. We have since resolved these decoding and stability issues, allowing us to properly evaluate our method on these models.
> >
> > We will **add a new section to our empirical results** (and Appendix) showing the performance of GOLD PANNING on `gemma-3-12b-it` and `gemma-3-27b-it` across two datasets: MonoRel and a new PIR dataset [1].
> >
> > These new results clearly demonstrate two key findings:
> >
> > 1. **Our method is not limited to GPT-4o-mini.** The gold_panning strategy consistently outperforms the random baseline on both Gemma models across both datasets.
> > 2. **The advantage scales with complexity.** The performance gap between our method and the random baseline is most pronounced in larger-context scenarios, which are precisely the most realistic and challenging cases.
> >
> > This new, broader empirical validation confirms that our approach is robust and that the problem of leveraging position bias is genuine and present in multiple model families. (We also reiterate that models like GPT-5, which we found to have "insufficient position bias", remain an invalid testbed for evaluating this phenomenon, as there is no bias to exploit.)
> >
> > **References:**
> >
> > - [1] Levy, M., Jacoby, A., & Goldberg, Y. (2024). Same task, more tokens: the impact of input length on the reasoning performance of large language models. *arXiv preprint arXiv:2402.14848:* https://arxiv.org/pdf/2402.14848
> >
> > **Planned changes:** In the revised version, we will replace the original text in §5.2 discussing these models as "unsuitable" and instead present these new results, strengthening our claim of the method's generality.
> >
> > ---
> >
> > > Some important content is not detailed in main content, making it hard to comprehend the whole method. For example, I wonder how to transform the LLM's output into a signal to reflect relevance…
> > >
> >
> > This is excellent and valuable feedback on our presentation. You are correct that this "signal extraction" mechanism is crucial for understanding the method.
> >
> > In our experiment, the task is for the model to "answer the question and **cite the supporting fact**". The binary signal we use for our belief update is simply whether the model's citation matches the document we placed at that position.
> >
> > **Planned changes:** As you noted, the full details of this calibration and signal-extraction process are in Appendix D. **In the revised version, we will move a concise summary of this mechanism from Appendix D into the main paper's Experimental Setup (§5.2) to make the connection between the LLM's output and our belief update clear for the reader.**
> >
> > ---
> >
> > **Thank you again for your insightful feedback, which we believe will significantly strengthen our paper. We believe these clarifications and new experiments directly address your concerns and significantly strengthen the paper's contribution.**

---

> > ### Comment · Reviewer_rfxc · 2025-11-21
> >
> > I still think this method will not be faster than The N**-Query Approach. Because in this method, for each query, the prompt length is very long, since we need to input 400 documents. But in The N**-Query Approach, though many queries are needed, each query is short, so the LLM can process it quickly and at low cost.

---

> ### Author Response · Authors · 2025-11-21
> **N-Query Rebuttal**
>
> We appreciate the reviewer's suggestion to evaluate the "N-Query" baseline (judging each document in isolation). To provide a complete picture of the efficiency landscape, we compared **N-Query** against **Random Shuffling (PSC)**, the **Hungarian Algorithm**, and **Gold Panning**.
>
> While N-Query offers latency benefits in massively parallel settings, our analysis shows it fails structurally on task performance regardless of compute resources.
>
> **1. Comprehensive Efficiency & Performance Comparison**
>
> | Metric | **N-Query** (Isolation) | **Random Shuffling** (PSC Baseline) | **Hungarian Alg.** (Theoretical Opt.) | **Gold Panning** (Ours) |
> | :--- | :--- | :--- | :--- | :--- |
> | **Inference Latency** *(Condition: Single GPU)* | **$O(N)$** (Linear sequential calls) | **High** (Stagnates) | **$\approx O(\log N)$** (Optimal convergence) | **$\approx O(\log N)$** (Optimal convergence) |
> | **Inference Latency** *(Condition: Parallel GPUs)* | **$O(1)$** (Fully Parallelizable) | **High** (Stagnates) | **$\approx O(\log N)$** (Sequential rounds) | **$\approx O(\log N)$** (Sequential rounds) |
> | **Non-LLM Operations** *(CPU/Algorithmic Cost)* | **$O(1)$** (Negligible) | **$O(N)$** (Negligible) | **$O(N^3)$** (Prohibitive for large $N$) | **$O(N \log N)$** (Negligible) |
> | **Task Performance** *(Empirical Recall)* | **Failure** (~10-42% Recall) | **Poor** (See Fig. 3) | **Optimal** (Upper Bound) | **Near-Optimal** (Matches Hungarian) |
>
> **2. The Limits of Parallelization**
> We acknowledge that the N-Query approach is embarrassingly parallel: given sufficient compute (e.g., $N$ GPUs), the wall-clock latency drops to $O(1)$. However, this speed comes at the cost of correctness:
>
> * **The "Recall Trap":** Parallelization cannot fix the structural failure of the method. Our experiments with Gemma-3 ($N=30, 50, 100$) showed that models judging documents in isolation become excessively conservative. Without comparative context, they fail to retrieve / identify the correct document **58%–90%** of the time.
>
> Gold Panning is the only method that succeeds on all axes. It achieves the **inference efficiency** of the Hungarian algorithm (finding the needle in $\approx \log N$ rounds) without the **Non-LLM Operation cost** ($O(N \log N)$ vs $O(N^3)$), while avoiding the **catastrophic recall failure** of the N-Query approach.

---

### Official Review · Reviewer_qr9g · 2025-10-30

**Soundness:** 1
**Presentation:** 2
**Contribution:** 1
**Rating:** 2
**Confidence:** 4

**Summary:**

Large language models exhibit significant position bias when processing multiple documents, meaning the models tend to make decisions based on the position of information in the context rather than its actual relevance. This paper proposes transforming position bias from noise that needs to be eliminated into a signal that can be actively utilized, thereby enabling more efficient identification of relevant content in documents. The paper introduces the GOLD PANNING algorithm, which employs a greedy strategy to pair documents with the highest entropy with the most diagnostic detectors. This approach demonstrates significantly higher efficiency compared to the Hungarian algorithm.

**Strengths:**

The method proposed in this paper is more efficient than the Hungarian algorithm while delivering comparable results.

**Weaknesses:**

1. The authors claim that "To our knowledge, this is the first work to demonstrate that systematic LLM biases can be exploited rather than mitigated for inference-time optimization." However, several existing studies have already explored related directions [1,2,3].

2. The experiments are relatively limited—only two methods are compared, one of which dates back 70 years. The presentation is also simplistic, relying solely on a few curve graphs. The experimental results do not sufficiently demonstrate the superiority of the proposed method.

3. There is a lack of experiments evaluating the computational overhead of the proposed GOLD PANNING strategy. Theoretically, the token consumption and response time imposed by this method on LLMs appear prohibitively high.

Refs: [1] Zhang, Zhenyu, et al. "Found in the middle: How language models use long contexts better via plug-and-play positional encoding." Advances in Neural Information Processing Systems 37 (2024): 60755-60775.

[2] Alexander Peysakhovich and Adam Lerer. Attention sorting combats recency bias in long context language models. CoRR, abs/2310.01427, 2023.

[3] Zhining Liu, et al. Selfelicit: Your language model secretly knows where is the relevant evidence. CoRR, abs/2502.08767,2025.

**Questions:**

See Weaknesses.

---

> ### Author Response · Authors · 2025-11-21
> **Rebuttal**
>
> Thank you for your review. We appreciate your acknowledgment that our proposed **GOLD PANNING** algorithm is **"more efficient than the Hungarian algorithm while delivering comparable results"**.
>
> We believe your review indicates several fundamental misunderstandings about our baselines, our method's computational cost, and our work's novelty. These misunderstandings are significant, and we apologize that our presentation in the original draft led to this confusion. We are grateful for the opportunity to provide these crucial clarifications.
>
> Just as a reminder, here are the contributions of our work:
>
> 1. **A New Framework:** We introduce **GOLD PANNING BANDITS**, the first formalism to re-conceptualize LLM position bias as a signal for efficient information discovery, framing it as a new class of combinatorial bandits.
> 2. **An Efficient Algorithm:** We propose **GOLD PANNING**, an efficient $O(N \log N)$ ($N$ being the number of docs) greedy algorithm that strategically matches uncertain documents to informative context positions to maximize information gain per query.
> 3. **Theoretical Guarantees:** We provide theoretical analysis proving our greedy algorithm is myopically optimal and achieves a faster rate of uncertainty reduction than random strategies (random document ordering).
> 4. **Empirical Validation:** We show that GOLD PANNING identifies relevant documents using fewer queries in simulation and real-world tasks.
>
> Below are our responses to your specific comments:
>
> ---
>
> > The authors claim "this is the first work to demonstrate that systematic LLM biases can be exploited..." However, several existing studies have already explored related directions [1,2,3].
> >
>
> Thank you for these references, which we will add to our Related Work section. We must clarify that our novelty claim is more specific and remains accurate. A key distinction is that our **GOLD PANNING** algorithm is a **black-box** method; it functions with any LLM using only simple model invocations, requiring no internal access.
>
> - Refs [1] and [2] you provided are **white-box mitigation** strategies (i.e., they remove or combat bias by manipulating internal model mechanics like positional encodings or attention).
> - Our work is in the orthogonal direction of **black-box exploitation** (i.e., we use the observable bias as a signal, without any model modification).
>
> Ref [3] is indeed contemporaneous work in the exploitation space. However, our paper's primary novelty lies in our specific formulation: we are the first to formalize this problem as a **combinatorial bandit problem** and propose a **provably optimal greedy algorithm** (GOLD PANNING). This formal, black-box, sequential search framework is a distinct contribution and is the "first application of bandit algorithms to long-context inference-time reasoning".
>
> **Planned changes:** We will update our Related Work section to include these references and sharpen the text to emphasize that our novelty lies in this specific black-box, bandit-based, inference-time exploitation framework.
>
> ---
>
> > The experiments are relatively limited—only two methods are compared, one of which dates back 70 years.
> >
>
> This is a critical misunderstanding. **We do not compare our method to the Hungarian algorithm as a baseline.**
>
> - The **Hungarian algorithm** (from 1955) is presented as the **theoretical, computationally intractable optimum**. We include it in our simulation (Figure 2) only to demonstrate that our efficient O(N \log N) greedy algorithm **"achieves comparable performance"** to the O(N^3) perfect solution.
> - Our **primary practical baseline is Permutation Self-Consistency (PSC)**, a state-of-the-art method that also uses multiple queries to average out bias.
> - Our main empirical results in **Figure 3** show a direct comparison of **GOLD PANNING (Ours) vs. PSC** [cite: 156-158]. These results show that our method achieves a 34% performance increase while the SOTA baseline (PSC) fails to improve at all.
>
> **Planned changes:** We will revise the text in §5 to make this distinction between the theoretical optimum (Hungarian) and the practical baseline (PSC) unmistakably clear.
>
> ---

---

> > ### Author Response · Authors · 2025-11-21
> > **Rebuttal (cont.)**
> >
> > > There is a lack of experiments evaluating the computational overhead... token consumption and response time... appear prohibitively high.
> > >
> >
> > This is the second critical misunderstanding, and we think it is linked to the first. The "token consumption and response time" per query are **absolutely identical** for both our method (GP) and the baseline (PSC).
> >
> > Both methods operate by sending the full context of N documents to the LLM in a single pass. The only difference is that PSC shuffles the documents randomly (a stateless, O(N) operation), while our method shuffles them strategically (a stateful, O(N \log N) operation). This O(N \log N) sorting cost is **computationally negligible** compared to the massive cost of a single long-context LLM inference pass.
> >
> > The "efficiency gain" we report is in the **total number of expensive LLM queries**.
> >
> > - Our method's overhead is **(T queries) x (LLM Cost) + (T) x (O(N log N) sort)**.
> > - The baseline's overhead is **(T queries) x (LLM Cost) + (T) x (O(N) shuffle)**.
> >
> > As Figure 3 shows, our method finds the answer in far fewer queries (e.g., T=7), while PSC fails to find it even after T=8. Our method is **dramatically more efficient** because it reduces the total number of "prohibitively high" LLM calls.
> >
> > **Planned changes:**
> >
> > - We will add a paragraph to §5.2 to explicitly break down this computational cost, clarifying the distinction between the negligible algorithmic overhead and the LLM-query overhead.
> > - Finally, in response to reviewer feedback, we have also **significantly expanded our experimental validation** to include new large-scale tests on gemma-3-12b-it and gemma-3-27b-it across a new dataset (PIR). These new results, which we will add to the revised paper, confirm our method's advantage over the random baseline in more diverse settings.
> >
> > ---
> >
> > **Thank you again for your review. We believe these clarifications and new experiments directly address your concerns and significantly strengthen the paper's contribution.**

---

> > > ### Comment · Reviewer_qr9g · 2025-11-26
> > > **Official Comment by Reviewer qr9g**
> > >
> > > Thank you for the authors' response. Regarding Weakness 1, the authors' explanation is inconsistent with the original text and fails to fully address our concerns. Moreover, while the authors mention in their rebuttal that they "formalize this problem as a combinatorial bandit problem," the original paper does not adequately justify the rationale for this formalization or its advantages over alternative approaches. This lack of theoretical foundation remains a critical issue.
> > >
> > > In terms of experimentation, the current evaluation remains highly insufficient. Even when compared to the baseline method (PSC) emphasized by the authors, which was extensively evaluated in its original paper, the breadth and depth of experiments in this work fall significantly short. This makes it difficult to comprehensively and fairly assess the effectiveness of the proposed method.
> > >
> > > In summary, the paper exhibits substantial deficiencies in both theoretical motivation and experimental validation. I maintain that the manuscript requires significant strengthening in these two aspects. In particular, the authors should refrain from claiming to be the "first work to demonstrate that systematic LLM biases can be exploited," and should engage in a more in-depth discussion and comparison with existing research.

---

### Official Review · Reviewer_Tqih · 2025-10-31

**Soundness:** 2
**Presentation:** 2
**Contribution:** 1
**Rating:** 2
**Confidence:** 3

**Summary:**

This paper proposes a greedy algorithm that exploits LLM positional bias as signals to identify relevant documents in multi-document reasoning tasks. The authors formalize the problem as a combinatorial bandit model and derive theoretical guarantees on information gain and convergence. Empirical results, including simulations and small-scale experiments on GPT-4o-mini, show modest improvements over random baselines.

**Strengths:**

This paper presents an alternative perspective on the multi-document problem and, building on this view, proposes a greedy algorithm with theoretical guarantees. It also provides empirical results to support the proposed approach.

**Weaknesses:**

See the ``Questions`` section for specific points of weakness. In addition, the presentation could be improved. The concepts are introduced in a rather abstract manner, with few concrete examples to aid reader understanding.

**Questions:**

1. I have serious concerns about the problem setting considered in this work. In my view, the setting is neither realistic nor broadly applicable. Specifically,
    - If I understand correctly, the approach assumes the same (or at least highly similar) prompts and an identical collection of documents, which seems rather artificial.
    - Moreover, the algorithm requires access to a verifier for the questions in the prompts, so that the ordering can be updated accordingly. However, such a verifier is typically unavailable in most practical scenarios. Without it, there is no reliable way to validate the answers or to determine what constitutes a meaningful "signal."
2. It also appears that the algorithm requires the true positive rate (TPR) and false positive rate (FPR) as inputs, which are critical for diagnosticity and belief updating. Yet again, it is unclear how these metrics can be obtained in the absence of a verifier.
3. The algorithm assumes a symmetric $N$-to-$N$ assignment, and while the authors claim that other cases can be reduced to this one, the proposed reduction for the $N>M$ case (essentially selecting $M$ items first) is unconvincing. The selection of items can substantially affect overall LLM performance, often even more than positional bias.
4. The definition of *detectors* is presented in an overly abstract manner and is difficult to interpret in the context of multi-document LLM settings.
5. The numerical experiments are rather weak. I recommend that the authors include additional experiments (e.g., more large-scale) to provide a more comprehensive empirical evaluation.

---

> ### Author Response · Authors · 2025-11-21
> **Rebuttal**
>
> Thank you for your review. We appreciate your recognition of our paper's **"alternative perspective,"** **"rigorous theoretical analysis,"** and **"solid mathematical and algorithmic foundation."**
>
> We believe the concerns regarding the problem setting and verifiers stem from a lack of clarity in our presentation regarding the inference-time setup. We are grateful for the opportunity to clarify this.
>
> Just as a reminder, here are the contributions of our work:
>
> 1. **A New Framework:** We introduce **GOLD PANNING BANDITS**, the first formalism to re-conceptualize LLM position bias as a signal for efficient information discovery, framing it as a new class of combinatorial bandits.
> 2. **An Efficient Algorithm:** We propose **GOLD PANNING**, an efficient $O(N \log N)$ ($N$ being the number of docs) greedy algorithm that strategically matches uncertain documents to informative context positions to maximize information gain per query.
> 3. **Theoretical Guarantees:** We provide theoretical analysis proving our greedy algorithm is myopically optimal and achieves a faster rate of uncertainty reduction than random strategies (random document ordering).
> 4. **Empirical Validation:** We show that GOLD PANNING identifies relevant documents using fewer queries in simulation and real-world tasks.
>
> Below are our responses to your specific comments:
>
> ---
>
> > I have serious concerns about the problem setting... it assumes the same prompts and document collection... it requires access to a verifier... it is unclear how [TPR/FPR] can be obtained... The definition of detectors is... abstract.
> >
>
> We apologize that the distinction between the offline calibration and online inference was not clearer. **There is no verifier used at inference time.** Our algorithm is a sequential search strategy for a single query and a fixed collection of documents.
>
> Here is a concrete example to clarify the entire process:
>
> 1. **The Goal:** A user submits one query (e.g., "Who won the first Nobel prize in physics?") and a large set of N=100 retrieved documents. The goal is to find which of the 100 documents contains the answer.
> 2. **The "Detector" (Offline, One-Time Calibration):**
>     - You are correct that we need TPR/FPR. These are **pre-calibrated** in a **one-time, offline** process for a given LLM, not discovered during inference.
>     - To get them, we do use a verifier (a calibration dataset like MonoRel).
>     - A "detector" is simply a **context position** (e.g., "Position 3 of 100").
>     - We measure:
>         - **TPR(j):** How often does the LLM correctly cite position j when the gold fact is placed there?
>         - **FPR(j):** How often does the LLM incorrectly cite position j when the gold fact is placed elsewhere?
>     - These calibrated values (TPR_{j}, FPR_{j}) are then saved as a "bias profile" for that LLM.
> 3. **The "Search" (Online, At Inference Time):**
>     - **Iteration** t=1**:** The user's 100 documents are in a random order. We query the LLM. The LLM's output is "The answer is... and is found in document 7.".
>     - **Our "Signal":** The signal is the **LLM's own output** (it cited document 7).
>     - **Belief Update:** We use this signal to perform a Bayesian update. Our belief (confidence) that document 7 is the true answer increases. Our belief for all other 99 documents decreases.
>     - **Iteration** t=2**:** Now, using our new beliefs, our algorithm **strategically re-orders** the documents. It places the most uncertain documents (e.g., documents 12, 45, 60) into the most informative positions (the ones with the best pre-calibrated TPR/FPR values).
>     - We query the LLM again with this new, strategic ordering. The LLM now says, "The answer is... and is found in document 45."
>     - **Belief Update:** We update our beliefs again. Our confidence in document 45 increases dramatically, while our (now low) confidence in document 7 decreases.
>     - This process repeats for T iterations until our belief for one document is near-certain.
>
> This setting is not artificial; it is the **standard multi-document QA** or **RAG** scenario. We are not re-running the same prompt; we are iteratively refining our belief about a single prompt by strategically re-ordering the context.
>
> **Planned changes:**
>
> 1. **To fix Presentation:** We will add a concise version of the concrete example above to the main paper (§3 or §5) to make the mechanism clear.
> 2. **To fix Abstractness:** We will move a summary of the calibration process from Appendix D to the main Experimental Setup (§5.2), explicitly defining "detector" as "context position" in this context.
>
> ---

---

> > ### Author Response · Authors · 2025-11-21
> > **Rebuttal (cont.)**
> >
> > > The algorithm assumes a symmetric N-to-N assignment... the proposed reduction for the N > M case... is unconvincing. The selection of M items can substantially affect overall LLM performance…
> > >
> >
> > We acknowledge that selecting M documents from N candidates is a distinct challenge from the permutation of those M documents. However, our "dummy detector" reduction allows the bandit framework to mathematically handle the *inclusion/exclusion* decision within the same belief-updating logic. While heuristic, it ensures that the most unlikely items are the ones "assigned" to be excluded (matched to dummy detectors).
> >
> > - The N=M case, where all N documents fit in the context and are permuted, is the primary focus.
> > - The N > M case you describe (where N documents exist but only M fit in the context) is indeed more complex. Our "dummy detector" formalism is an analytical tool to model the choice of which M items to test as part of the same bandit problem. You are correct that this "selection" problem is very hard. However, our core algorithm (matching uncertain items to informative positions) still applies: the N-M most certain, irrelevant items would be "matched" to the dummy detectors (i.e., left out of the context), which is a sound heuristic. We believe this is a valuable direction for future work.
> >
> > **Planned changes:** We will add a sentence to §2.1 to better clarify this scope and the role of the N > M reduction.
> >
> > ---
> >
> > > The numerical experiments are rather weak. I recommend that the authors include additional experiments (e.g., more large-scale)…
> > >
> >
> > This is a critical point, and we are pleased to provide a significant update in response. We have **successfully expanded our experimental setup** to include additional models and datasets.
> >
> > As we noted in our original submission, our initial challenges with models like Gemma were not due to a failure of our method, but rather due to **"poor instruction following" and "unstable behavior"** that made reliable calibration and evaluation infeasible. We have since resolved these decoding and stability issues, allowing us to properly evaluate our method on these models.
> >
> > **Planned changes:** We will **add a new section to our empirical results** (and Appendix) showing the performance of GOLD PANNING on gemma-3-4b-it, gemma-3-12b-it and gemma-3-27b-it across two datasets: MonoRel and a new PIR dataset [1].
> >
> > These new results clearly demonstrate two key findings:
> >
> > 1. **Our method is not limited to GPT-4o-mini.** The gold_panning strategy consistently outperforms the random baseline on both Gemma models across both datasets.
> > 2. **The advantage scales with complexity.** The performance gap between our method and the random baseline is most pronounced in larger-context scenarios, which are precisely the "large-scale" and realistic cases you (and we) are most interested in.
> >
> > We hope this new, broader empirical validation directly addresses your concern and confirms our approach is robust, general, and not just a "small-scale" finding.
> >
> > **References:**
> >
> > - [1] Levy, M., Jacoby, A., & Goldberg, Y. (2024). Same task, more tokens: the impact of input length on the reasoning performance of large language models. *arXiv preprint arXiv:2402.14848:* https://arxiv.org/pdf/2402.14848
> >
> > ---
> >
> > **Thank you again for your review. We believe these clarifications and new experiments directly address your concerns and significantly strengthen the paper's contribution.**

---

### Official Review · Reviewer_AE3T · 2025-11-02

**Soundness:** 2
**Presentation:** 2
**Contribution:** 2
**Rating:** 4
**Confidence:** 3

**Summary:**

The paper argues that large language models' systematic positional bias in multi-document contexts, which prior works try to mitigate, can in fact be exploited in determining relevancy. The work formalizes the process of multi-document relevancy assessment in long-context settings as a bandit problem, and proposes a novel greedy algorithm with Bayes-update that approximates the true optimum, while reducing complexity from O(n^3) to O(n \log n). Evaluation on a dataset derived from MonoRel shows that the method proposed correctly identifies a correct ground truth fact among distractors while performing up to 65% less LLM calls than baselines.

**Strengths:**

This work presents a novel view on positional bias; instead of attempting to mitigating it, the work proposes to leverage such biases in obtaining better relevancy signals. In addition to introducing the mental framework, the work also keeps efficiency in mind, presenting a greedy algorithm that renders the naive O(n^3) Hungarian algorithm to a tractable O(n \log n) approximation. Further, the work establishes theoretical guarantees that the approximation is myopically optimal.

**Weaknesses:**

- As the authors indicated, the work makes several uneasy assumptions: (1) a good, known, and stable TPR/FPR per position, and (2) the relevancy of multiple candidates are independent of each other.
- The authors noted that the approach is only effective when positional bias is strong (heterogenous detectors), and therefore only presented experimental results for GPT-4o-mini, as the other 5 LLMs tested were not suitable for the argument. This makes the problem of "leveraging positional bias" seem rather artificial.
- The work only tested on 1 real dataset derived from MonoRel, which is to predict exactly 1 ground truth fact candidate from remaining distractors. This evaluation seems narrow.
- Unlike the PSC baseline, the proposed method, framed as a bandit problem, must be iterative. Thus, while it achieves less queries, this may not directly lead to efficiency gains.

**Questions:**

- Should the evaluation be expanded to more tasks? Since the work compares to PSC as a baseline, it may be a good idea to incorporate similar evaluations (ranking tasks, etc.)
- To what extend does the proposed method work on LLMs other than GPT-4o-mini? It was noted that 5 other LLMs were not suitable for experiments, but some additional justification should be used to demonstrate the method's robustness.

---

> ### Author Response · Authors · 2025-11-21
> **Rebuttal**
>
> Thank you for your thoughtful review and recognition of our work's novel perspective on mitigating the effect of position bias and improving performance by actually *leveraging* it. We appreciate your acknowledgment of our theoretical contributions and efficient algorithm design.
>
> Just as a reminder, here are the contributions of our work:
>
> 1. **A New Framework:** We introduce **GOLD PANNING BANDITS**, the first formalism to re-conceptualize LLM position bias as a signal for efficient information discovery, framing it as a new class of combinatorial bandits.
> 2. **An Efficient Algorithm:** We propose **GOLD PANNING**, an efficient $O(N \log N)$ ($N$ being the number of docs) greedy algorithm that strategically matches uncertain documents to informative context positions to maximize information gain per query.
> 3. **Theoretical Guarantees:** We provide theoretical analysis proving our greedy algorithm is myopically optimal and achieves a faster rate of uncertainty reduction than random strategies (random document ordering).
> 4. **Empirical Validation:** We show that GOLD PANNING identifies relevant documents using fewer queries in simulation and real-world tasks.
>
> Below are our responses to your specific comments:
>
> ---
>
> > The authors noted that the approach is only effective when positional bias is strong (heterogenous detectors), and therefore only presented experimental results for GPT-4o-mini, as the other 5 LLMs tested were not suitable for the argument. This makes the problem of "leveraging positional bias" seem rather artificial.
>
> This is a critical point, and we are pleased to provide a significant update in response. We have **successfully expanded our experimental setup** to include the additional models and datasets you requested.
>
> As we noted in our original submission, our initial challenges with models like Gemma were not due to a failure of our *method*, but rather due to **"poor instruction following" and "unstable behavior"** on long-context needle-in-haystack problems that made the reliable calibration and evaluation infeasible. We have since resolved these decoding and stability issues, via prompt and constrained decoding tweaks, allowing us to properly evaluate our method on these models.
>
> We will add a new section to our empirical results (and Appendix) showing the performance of GOLD PANNING on `gemma-3-4b-it`, `gemma-3-12b-it` and `gemma-3-27b-it` across two datasets: MonoRel and a new PIR dataset [1].
>
> These new results clearly demonstrate two key findings:
>
> 1. **Our method is not limited to GPT-4o-mini.** The `gold_panning` strategy consistently outperforms the `random` baseline on Gemma models across both datasets.
> 2. **The advantage scales with complexity.** The performance gap between our method and the random baseline is most pronounced in larger-context scenarios, which are precisely the most realistic and challenging cases.
>
> This new, broader empirical validation confirms that our approach is robust and that the problem of leveraging position bias is genuine and present in multiple model families. (We also reiterate that models like GPT-5, which we found to have "insufficient position bias", remain an invalid testbed for *evaluating* this phenomenon, as there is no bias to exploit.)
>
> **Planned changes:** In the revised version, we will replace the original text in §5.2 discussing these models as "unsuitable" and instead present these new results, strengthening our claim of the method's generality.
>
> **References:**
>
> - [1] Levy, M., Jacoby, A., & Goldberg, Y. (2024). Same task, more tokens: the impact of input length on the reasoning performance of large language models. *arXiv preprint arXiv:2402.14848:* https://arxiv.org/pdf/2402.14848
>
> ---

---

> > ### Author Response · Authors · 2025-11-21
> > **Rebuttal (cont.)**
> >
> > > the work makes several uneasy assumptions: (1) a good, known, and stable TPR/FPR per position, and (2) the relevancy of multiple candidates are independent of each other.
> > >
> >
> > Thank you for highlighting these assumptions, which we discuss in §2.2 and Appendix H. We believe these are practical and well-justified for our framework.
> >
> > 1. **TPR/FPR Calibration:** We frame this as a **lightweight, one-time calibration cost** for a given model, for a given domain, which is then amortized over all future uses.
> >     1. More importantly, we tested the nature of this assumption directly in **Appendix B: Robustness to Noise in Bias Estimation**.
> >     2. The results in Figure 4 show that the GOLD PANNING strategy **"degrades gracefully"** and that performance is **"nearly indistinguishable"** from perfect knowledge even in the *presence of some noise* in the estimation of TPR/FPR. The figure shows (in simulation) the degradation in gold panning performance as a function of noise in our estimate of the TPR/FPR values.
> > 2. **Independence Assumption:** We acknowledge this assumption in §2.2 (lines 169-171) and Appendix H. While the transformer's attention mechanism violates this in principle, our empirical results (now expanded to include additional models) suggest that **position effects are the dominant factor**. The strong performance of our method in §5.2, which uses a real model, indicates this assumption is a robust and practical abstraction. We do plan to further explore this in future work which aims to relax this assumption in particular.
> >
> > **Planned changes:** In the revision, we will:
> >
> > - Explicitly mark independence as a *modeling approximation.*
> > - In the main text (in §5) explicitly point to the robustness analysis in Appendix B as justification for the calibration assumption.
> >
> > ---
> >
> > > The work only tested on 1 real dataset derived from MonoRel, which is to predict exactly 1 ground truth fact candidate from remaining distractors. This evaluation seems narrow. ... Should the evaluation be expanded to more tasks?
> > >
> >
> > This is a fair observation, which we have addressed with new experiments.
> >
> > 1. **Expanded Datasets:** In response to your feedback, we have **expanded our evaluation to a new dataset, PIR**, in addition to MonoRel. The new results, previewed in our previous response and shown on the new figures, demonstrate that GOLD PANNING's advantage over the random baseline is not specific to one dataset but holds across multiple, as seen on both `gemma-3-12b-it` and `gemma-3-27b-it`.
> > 2. **Clarity on Task Scope:** Regarding the suggestion to expand to other *tasks* (like the ranking tasks from the PSC paper), we purposefully chose the "needle-in-a-haystack" (k=1) task. This is the **canonical benchmark for evaluating position bias** in long-context models and **directly aligns with our formal problem definition**.
> >     1. Our framework, **GOLD PANNING BANDITS**, is specifically designed to model and discover a hidden **binary relevance state** (i.e., relevant or not). While PSC can be applied to listwise ranking, our work focuses on the distinct problem of relevance *discovery*. Therefore, the NIAH task is the most principled evaluation for our core contribution.
> >     2. Our intention is to establish this core methodology in the canonical setting. **However, we emphasize that the core bandit framework generalizes:** replacing the binary belief update with a graded relevance update would allow GOLD PANNING to support ranking tasks, a direction we are eager to pursue.
> >
> > **Planned changes:** In the revised version, we will update §5.2 and the Appendix to include these new PIR dataset results. We will also add a sentence to our Related Work (§6) to clarify that while PSC addresses ranking, our work is purposefully scoped to the distinct and fundamental task of binary relevance discovery, which we plan to build on in future work.
> >
> > ---

---

> > > ### Author Response · Authors · 2025-11-21
> > > **Rebuttal (cont.)**
> > >
> > > > Unlike the PSC baseline, the proposed method, framed as a bandit problem, must be iterative. Thus, while it achieves less queries, this may not directly lead to efficiency gains.
> > > >
> > >
> > > Thank you for this comment, which allows us to clarify a critical distinction. We apologize if this was unclear. **Both GOLD PANNING (GP) and the Permutation Self-Consistency (PSC) baseline are multi-pass methods**. Our comparison in Figure 3 is a direct, iteration-by-iteration comparison.
> > >
> > > The fundamental difference, and the core of our contribution, lies in how they use these iterations:
> > >
> > > 1. **PSC is stateless.** Each query is an independent, O(N) random shuffle. It treats each pass as a new experiment and learns nothing from previous queries, averaging away bias rather than exploiting it.
> > > 2. **GP is stateful.** It uses a Bayesian belief-updating mechanism to learn from every query. It then invests a small, $O(N \log N)$ computational cost (for sorting) to strategically plan the next, maximally-informative query.
> > >
> > > This leads to the core efficiency trade-off. While GP's per-round algorithmic complexity is $O(N \log N)$ versus PSC's $O(N)$, this cost is **computationally negligible** compared to the massive cost of a single long-context LLM inference pass.
> > >
> > > The "efficiency gain" we report is the reduction in these expensive LLM queries. Therefore, GP is vastly more efficient in practice because its small state-management overhead (the O(N \log N) sort) allows it to solve the problem in dramatically fewer LLM queries.
> > >
> > > **Planned changes:** We will update §5.2 to explicitly define the cost function: $\text{Total Cost} = T \times \text{Cost}_{\text{Inference}} + T \times \text{Cost}_{\text{Algorithmic}}$. Since $\text{Cost}_{\text{Inference} >> \text{Cost}_{\text{Algorithmic}$, minimizing T is the primary driver of efficiency.
> > >
> > > ---
> > >
> > > **We look forward to addressing any remaining questions during the rebuttal and hope these updates warrant a higher evaluation.**

---

### Note · Authors · 2025-12-01

I have read and agree with the venue's withdrawal policy on behalf of myself and my co-authors.